# Learning Positive Functions with Pseudo Mirror Descent

**Yingxiang Yang**[*]
UIUC
yyang172@illinois.edu

**Haoxiang Wang**
UIUC
hwang264@illinois.edu

**Negar Kiyavash**
EPFL
negar.kiyavash@epfl.ch

**Niao He**
UIUC
niaohe@illinois.edu

## Abstract

The nonparametric learning of positive-valued functions appears widely in machine learning, especially in the context of estimating intensity functions of point processes. Yet, existing approaches either require computing expensive projections or semidefinite relaxations, or lack convexity and theoretical guarantees after introducing nonlinear link functions. In this paper, we propose a novel algorithm, pseudo mirror descent, that performs efficient estimation of positive functions within a Hilbert space without expensive projections. The algorithm guarantees positivity by performing mirror descent with an appropriately selected Bregman divergence, and a pseudo-gradient is adopted to speed up the gradient evaluation procedure in practice. We analyze both asymptotic and nonasymptotic convergence of the algorithm. Through simulations, we show that pseudo mirror descent outperforms the state-of-the-art benchmarks for learning intensities of Poisson and multivariate Hawkes processes, in terms of both computational efficiency and accuracy.

## 1 Introduction

Learning positive-valued functions (or positive functions for short) in Hilbert spaces is pervasive in machine learning, especially when estimating intensity functions of point processes. In recent years, there has been a surge of interest and demand for modeling large-scale time-series and discrete event data using point processes. This is fueled by a wide spectrum of applications ranging from modeling financial activities [Embrechts et al., 2011], to modeling network diffusion such as in disease propagation [Yang and Zha, 2013] and spread of news on social networks [Farajtabar et al., 2015, 2017], to tracking and control of large-scale and real-time systems [Craciun et al., 2015]. Despite this, progress has been slow on nonparametric learning of positive functions (or positive intensities in case of point processes).

### 1.1 Learning Positive Functions: Existing Results

**Semi-infinite/Semi-definite relaxations.** In regularized empirical risk minimization over a reproducing kernel Hilbert space (RKHS), the representer theorem [Schölkopf et al., 2001] allows one to write the estimate as a linear combination of reproducing kernels. Therefore, the optimization

---

[*]This work was supported in part by MURI grant ARMY W911NF-15-1-0479, ONR grant W911NF-15-1-0479, NSF CCF-1755829 and NSF CMMI-1761699.

problem reduces to a special instance of semi-infinite programming (SIP), which can then be solved using a variety of methods, such as cutting plane methods [Wu and Fang, 1999, Betrò, 2004, Kortanek and No, 1993, Papp, 2017]. If the RKHS has a polynomial kernel [Prestel and Delzell, 2013, Bagnell and Farahmand, 2015], then the problem further reduces to a sum-of-squares (SOS) optimization, which can be solved using semi-definite programming (SDP) solvers, e.g., Grant and Boyd [2014]. Although these approaches guarantee positivity, they are often limited to the batch learning setting, and are computationally expensive, thus unsuitable for learning large or streaming data sets.

**Link functions.** Another approach for enforcing positivity is to perform a change of variable via a pointwise mapping, $h : \mathbb{R} \to \mathbb{R}_+$, known as a link function. Examples include $h(t) = t^2$ and $h(t) = \exp(t)$, as well as various types of activation functions in neural networks [Mei and Eisner, 2017, Xiao et al., 2017]. By introducing $h$, the original problem of learning over a constrained set of functions is effectively transformed into an unconstrained one. Such methods have been successfully applied to nonparametric learning of intensity functions of Poisson and multivariate Hawkes processes [Flaxman et al., 2017, Yang et al., 2017]. However, despite their numerical advantage, the introduction of a link function often breaks convexity of the underlying learning problem. Consequently, the numerical results are not backed by theoretical guarantees.

**Projection.** When applying iterative optimization algorithms such as the gradient descent, an ad hoc way to enforce positivity of the intermediate updates is to perform projection. In a parametric setting, this can be carried out by solving a quadratic program (QP), with positivity constraint enforced on a large but finite set of points over the support of the estimate (see Appendix K for details). However, this approach does not guarantee an optimal solution due to relaxation of the constraints.

## 1.2 Our Contribution

Despite recent advances in learning positive functions, nonparametric learning algorithms that are both computationally efficient and provide theoretical guarantees remain largely elusive. In this paper, we design a pseudo mirror descent algorithm that leverages the classical mirror descent algorithm and a sequence of pseudo-gradients to achieve these goals. When the objective is smooth and the pseudo-gradient is close to the true gradient, we prove that the gradient norm vanishes at the rate of $\mathcal{O}(1/\sqrt{k})$, where $k$ is the number of iterations. Under a generalized version of the Polyak-Łojasiewicz condition [Karimi et al., 2016], we further show that the objective value converges to the optimal at the rate of $\mathcal{O}(1/k)$. For several point processes estimation applications of interest, including learning intensities of nonhomogeneous Poisson processes and multivariate Hawkes processes, we construct pseudo-gradients based on kernel embeddings, as the true functional gradients for these problems are not accessible in practice. We also conduct extensive numerical experiments on both synthetic and real-world datasets. Those numerical results show that pseudo mirror descent outperforms existing nonparametric approaches in terms of both efficiency and accuracy.

## 2 Learning Positive Functions in Hilbert Spaces

We first focus on a general optimization problem:

$$\min_{x \in \mathcal{H}_+} f(x), \tag{1}$$

where $\mathcal{H}$ is a Hilbert space that consists of functions mapping a compact support $\Omega \subset \mathbb{R}^d$ to $\mathbb{R}$, and $\mathcal{H}_+ := \{x \in \mathcal{H} : x(t) \geq 0, \ \forall t \in \Omega\}$. The topological dual of $\mathcal{H}$, which consists of continuous linear operators on $\mathcal{H}$, is denoted by $\mathcal{H}^*$, and the norm and inner product of $\mathcal{H}$ are denoted by $\|\cdot\|$ and $\langle\cdot,\cdot\rangle$, respectively. Next, we introduce notations and definitions that will be used frequently in our analysis.

**Functional gradient.** For a Gâteaux differentiable functional $f : \mathcal{H} \to \mathbb{R}$, denote its Gâteaux derivative by $[\mathrm{D}f(x)](\cdot)$. The functional gradient of $f$ at $x$, denoted by $\nabla f(x)$, belongs to $\mathcal{H}$ and satisfies $[\mathrm{D}f(x)](y) = \langle \nabla f(x), y \rangle$ for any $y \in \mathcal{H}$. By the Riesz representation theorem, $\nabla f(x)$ exists and is unique. Likewise, if $f$ is twice Gâteaux differentiable, one can define the Hessian of $f$

at $x$ by $\nabla^2 f(x) \in \mathcal{H}^*$, such that for any $y, z \in \mathcal{H}$, $[\mathrm{D}[\mathrm{D}f(x)](y)](z) = \langle z, [\nabla^2 f(x)](y) \rangle$. For more details, please see Bauschke and Combettes [2011].

**Bregman divergence and Fenchel conjugate.** Let $\mathrm{int}(\mathcal{H}_+)$ be the interior of $\mathcal{H}_+$, and consider a continuously differentiable functional $\Phi : \mathrm{int}(\mathcal{H}_+) \to \mathbb{R}$ that is $\mu$-strongly-convex with respect to some norm $\|\cdot\|_\sharp$. That is, $\Phi(x) \geq \Phi(y) + \langle \nabla\Phi(y), x-y \rangle + \frac{\mu}{2}\|x-y\|_\sharp^2$, $\forall x, y \in \mathrm{int}(\mathcal{H}_+)$. Define the Bregman divergence induced by $\Phi$ as $\Delta_\Phi(x, y) = \Phi(x) - \Phi(y) - \langle \nabla\Phi(y), x-y \rangle$ for $x, y \in \mathrm{int}(\mathcal{H}_+)$, and $\Delta_\Phi(x, y) \geq \frac{\mu}{2}\|x-y\|_\sharp^2$. The Fenchel conjugate of $\Phi$ is $\Phi^*(u) = \sup_{x \in \mathcal{H}}\{\langle x, u \rangle - \Phi(x)\}$, which is $\mu^{-1}$-Lipschitz-smooth with respect to $\|\cdot\|_{\sharp,*}$, which stands for the dual norm of $\|\cdot\|_\sharp$. [2]

In this paper, we aim at leveraging the classic mirror descent algorithm [Nemirovski and Yudin, 1983] to guarantee positivity. This approach requires the following assumption.

**Assumption 1.** *Suppose $\min_{x \in \mathcal{H}_+} f(x) = f^* > -\infty$ is achieved at $x^* \in \mathrm{int}(\mathcal{H}_+)$, and there exists a $\Phi : \mathrm{int}(\mathcal{H}_+) \to \mathbb{R}$ continuously differentiable and $\mu$-strongly-convex with respect to $\|\cdot\|_\sharp$, such that $\nabla\Phi^*(x) \in \mathrm{int}(\mathcal{H}_+)$ for $x \in \mathcal{H}$. Moreover, let $f_\Phi(x) = (f \circ \nabla\Phi^*)(x) = f(\nabla\Phi^*(x))$. We assume $\nabla f_\Phi(\nabla\Phi(x)) \in \mathcal{H}$ for $x \in \mathrm{int}(\mathcal{H}_+)$, and that $f_\Phi$ is $M\mu^{-1}$-Lipschitz-smooth for constant $M$:*

$$\|\nabla f_\Phi(\nabla\Phi(y)) - \nabla f_\Phi(\nabla\Phi(x))\|_\sharp \leq M\mu^{-1}\|\nabla\Phi(y) - \nabla\Phi(x)\|_{\sharp,*}, \quad \forall x, y \in \mathrm{int}(\mathcal{H}_+). \quad (2)$$

When $\Phi(x) = \|x\|^2/2$, we have $\nabla\Phi^*(x) = \nabla\Phi(x) = x$ and $\nabla f_\Phi(\nabla\Phi(x)) = \nabla f(x)$. In this case, (2) reduces to the standard smoothness assumption on the objective $f$. For more general choices of $\Phi$, a sufficient condition for (2) is when $f$ is Lipschitz smooth and $\nabla^2\Phi$ has uniformly bounded eigenvalues over $\mathrm{int}(\mathcal{H}_+)$. However this is not necessary as we will show in Section 3.

Intuitively, Assumption 1 can be interpreted by introducing a "dual space" [Bubeck et al., 2015], $\mathcal{H}' = \{\nabla\Phi(x) : x \in \mathrm{int}(\mathcal{H}_+)\}$, which is connected to the primal space $\mathcal{H}_+$ through a pair of mappings: $\nabla\Phi : \mathrm{int}(\mathcal{H}_+) \to \mathcal{H}'$ and $\nabla\Phi^* : \mathcal{H}' \to \mathrm{int}(\mathcal{H}_+)$. Notice that, in the "dual space", the objective and its gradient become $f_\Phi(\nabla\Phi(x))$ and $\nabla f_\Phi(\nabla\Phi(x))$, respectively. Therefore, Assumption 1 assumes smoothness of the objective in the "dual space", where the dependence on $\Phi$ is incorporated into the Lipschitz constant. A more detailed illustration can be found in Appendix A.

## 2.1 Pseudo-gradients

In practice, the exact gradient can be costly to evaluate, store, or transmit; sometimes it may also lack desired properties such as continuity or smoothness. To circumvent of these challenges, a common practice is to use a rough direction as a substitute of the exact gradient in optimization algorithms. Examples include pseudo-gradients [Poljak and Tsypkin, 1973], the gradient sign [Goodfellow et al., 2015], ternary gradients [Wen et al., 2017], and quantized gradients [Wu et al., 2018]. Among them, the concept of pseudo-gradient is the most general, and the starting point of our algorithm design – using mirror descent to guarantee positivity – further motivates us to introduce a generalized notion of pseudo-gradient that is compatible with the Bregman divergence.

**Definition 1** (Pseudo-gradient). *Consider an iterative algorithm initialized at $x^{(0)}$ and with intermediate updates $x^{(1)}, \ldots, x^{(k)}$, where each $x^{(k)}$ is generated from some given rule $r(x^{(k-1)}, g^{(k)})$ with a random direction $g^{(k)} \in \mathcal{H}$. Let $\mathcal{F}^{(k)}$ be the minimum $\sigma$-algebra generated by $x^{(0)}, \ldots, x^{(k)}$. Then, a pseudo-gradient for $f$ at $x^{(k)}$ is a random element $g^{(k+1)} \in \mathcal{H}$ satisfying*

$$\langle \nabla f_\Phi(\nabla\Phi(x^{(k)})), \mathbb{E}[g^{(k+1)}|\mathcal{F}^{(k)}] \rangle \geq 0. \quad (3)$$

The notion of pseudo-gradient was originally introduced in Poljak and Tsypkin [1973] as a random element in $\mathcal{H}$ that has an acute angle with the true gradient: $\langle \mathbb{E}[g^{(k+1)}|\mathcal{F}^{(k)}], \nabla f(x^{(k)}) \rangle \geq 0$. This can be retained from Definition 1 by setting $\Phi(x) = \|x\|^2/2$. Under the intuition that guided us to raise Assumption 1, Definition 1 defines a pseudo-gradient to have an acute angle with the gradient in the "dual space". Below we give a few examples of pseudo-gradients.

**Example 1** (Stochastic gradients are pseudo-gradients). *Suppose $g^{(k)} \in \mathcal{H}$ is a stochastic gradient of $f$ at $x^{(k-1)}$: $\mathbb{E}[g^{(k)}|\mathcal{F}^{(k-1)}] = \nabla f(x^{(k-1)})$. Then $g^{(k)}$ is a pseudo-gradient of $f$ at $x^{(k-1)}$.*

**Algorithm 1** Pseudo Mirror Descent Algorithm
---
1: **Input:** number of iterations $T$; step sizes $\{\eta_k\}_{k=0}^{T}$; objective $f$; strongly convex function $\Phi$.
2: **Initialize** $x^{(0)} \in \text{int}(\mathcal{H}_+)$.
3: **for** $k = 1$ to $T$ **do**
4:     Compute pseudo-gradient $g^{(k)}$.
5:     $x^{(k)} = \text{argmin}_{x \in \text{int}(\mathcal{H}_+)} \left\{ f(x^{(k-1)}) + \langle g^{(k)}, x - x^{(k-1)} \rangle + \eta_{k-1}^{-1} \Delta_\Phi(x, x^{(k-1)}) \right\}$.
6: **end for**
7: **Output:** estimated function $x^{(T)}$.
---

The proof is in Appendix D. While stochastic gradients are pseudo-gradients, the converse is not true. As is the case with the following examples, pseudo-gradients can, and often turn out to be, biased.

**Example 2** (Kernel embeddings are pseudo-gradients). *Suppose $K(\cdot, \cdot) : \Omega \times \Omega \to \mathbb{R}$ is a symmetric positive definite kernel satisfying $\langle x, \langle K, x \rangle \rangle \geq 0$ for any $x \in \mathcal{H}$. Let $K_t = K(t, \cdot)$ then*

$$g^{(k)}(t) = \langle K_t, \nabla f_\Phi(\nabla \Phi(x^{(k-1)})) \rangle$$

*is a pseudo-gradient of $f$ at $x^{(k-1)}$.*

**Example 3** (The sign of the gradient is a pseudo-gradient for $\mathcal{H} = \mathcal{L}_2(\Omega)$). *For any $x \in \text{int}(\mathcal{H}_+)$,*

$$\langle \nabla f_\Phi(\nabla \Phi(x)), \text{sgn}(\nabla f_\Phi(\nabla \Phi(x))) \rangle = \int_\Omega |[\nabla f_\Phi(\nabla \Phi(x))](t)| \mathrm{d}t \geq 0.$$

### 2.2  Pseudo Mirror Descent: Algorithm and Theory

In this section, we introduce a new algorithm, *Pseudo Mirror Descent*, that integrates the stochastic mirror descent with pseudo-gradients. The stochastic mirror descent has been extensively studied and widely applied to solving constrained optimization problems: see, e.g., the seminal work by Nemirovski et al. [2009]. When it comes to the positivity constraint, the stochastic mirror descent algorithm, leveraging a properly chosen Bregman divergence, leads to a simple multiplicative update rule that preserves positivity, and reduces the runtime in practice.

The pseudo mirror descent algorithm is described in Algorithm 1, with the main iteration:

$$x^{(k)} = \underset{x \in \text{int}(\mathcal{H}_+)}{\text{argmin}} \left\{ f(x^{(k-1)}) + \langle g^{(k)}, x - x^{(k-1)} \rangle + \eta_{k-1}^{-1} \Delta_\Phi(x, x^{(k-1)}) \right\}, \tag{4}$$

where $\Delta_\Phi(\cdot, \cdot)$ is the Bregman divergence induced by $\Phi$. When $\nabla \Phi^*(x) \in \text{int}(\mathcal{H}_+)$ for $x \in \mathcal{H}$, $x^{(k)}$ has an explicit expression, as we show below (see Appendix E for proof).

**Lemma 2.** *Under Assumption 1, the solution of* (4) *reduces to*

$$x^{(k)} = \nabla \Phi^*(\nabla \Phi(x^{(k-1)}) - \eta_{k-1} g^{(k)}).$$

Below is an example that applies Lemma 2.

**Example 4** (The generalized $I$-divergence). *Let $\mathcal{H} = \mathcal{L}_2[0, 1]$, and $\Phi(x) \triangleq \langle x, \log(x) - 1 \rangle$. Then $\Delta_\Phi(x, y) = \langle x, \log(x) - \log(y) \rangle$, and* (4) *reduces to $x^{(k)} = x^{(k-1)} \exp\{-\eta_{k-1} g^{(k)}\}$.*

**Selection of the Bregman divergence.** Example 4 gave an example of Bregman divergence, but the choice of Bregman divergence is rather flexible, and can be designed in a more general fashion. In the context of learning positive functions, any distance-generating function $\Phi$ such that $\nabla \Phi^*$ preserves positivity would be sufficient. Intuitively, this means that one could start out by choosing an appropriate $\nabla \Phi^*$ and determine the corresponding $\Phi$ subsequently. Following this way of designing the Bregman divergence, a few more examples could be easily constructed, including using $\Phi(x) = -\int \log x(t) \mathrm{d}t$, which leads to the Itakura-Saito divergence, as well as $\Phi(x) = \int 0.4 x^{2.5}(t) \mathrm{d}t$.

Next, we provide both asymptotic and nonasymptotic convergence analysis for Algorithm 1. It is noteworthy that none of our results assume convexity of the objective. To the best of our knowledge, these are the first proven convergence results on mirror descent with pseudo-gradient updates.

**Convergence of a vanishing gradient.** First, we prove that the pseudo-gradient and the true gradient are asymptotically orthogonal.

**Theorem 3.** *Suppose Assumption 1 holds, and the step sizes in* (4) *satisfy $\eta_k \geq 0$, $\sum_{k=0}^{\infty} \eta_k = \infty$, and $\sum_{k=0}^{\infty} \eta_k^2 < \infty$. In addition, let $g^{(k)}$ satisfy*

$$\mathbb{E}[\|g^{(k)}\|_{\sharp,*}^2|\mathcal{F}^{(k-1)}] \leq \lambda_k + \rho\langle\nabla f_\Phi(\nabla\Phi(x^{(k-1)})), \mathbb{E}[g^{(k)}|\mathcal{F}^{(k-1)}]\rangle \tag{5}$$

*where the sequence $\lambda_k \geq 0$ satisfies $\sum_{k=0}^{\infty} \eta_k^2\lambda_{k+1} < \infty$, and $\rho$ is a positive constant. Then, with probability 1, $\lim_{k\to\infty} f(x^{(k)})$ exists and*

$$\liminf_{k\to\infty}\langle\nabla f_\Phi(\nabla\Phi(x^{(k-1)})), \mathbb{E}[g^{(k)}|\mathcal{F}^{(k-1)}]\rangle = 0.$$

The proof can be found in Appendix H. The above theorem requires a set of assumptions on the step sizes, as well as an upper bound on the pseudo-gradient's norm, which are standard assumptions in optimization literature [Bottou et al., 2018, Poljak and Tsypkin, 1973]. Under such assumptions, the pseudo-gradient and the gradient eventually become orthogonal to each other in probability. This implies that either the angle between the pseudo-gradient and the gradient becomes asymptotically perpendicular, or the norm of the pseudo-gradient converges to 0. Since in Algorithm 1, one has the freedom of designing the pseudo-gradient, we can immediately claim that, if (i) the pseudo-gradient is set to always have an acute angle with the true gradient, and (ii) the norm ratio between the pseudo-gradient and the gradient is lower bounded, then the norm of the gradient converges to 0. An example of this is given in the following corollary (see proof in Appendix G).

**Corollary 4.** *In Algorithm 1, suppose $\nabla^2\Phi$ is positive definite, and let $g^{(k)} = \nabla f_\Phi(\nabla\Phi(x^{(k-1)}))$ or $g^{(k)} = \nabla f(x^{(k-1)})$. Then, we have $\lim_{k\to\infty}\|\nabla f(x^{(k)})\| = 0$ in probability.*

Note that, if $\nabla f(x^{(k)})$'s values are uniformly continuous, then this would further imply convergence towards a stationary point of the objective $f$.

Next, we investigate the nonasymptotic convergence rate of Algorithm 1 to characterize the behavior of the approximate solution for a finite number of iterations (see proof in Appendix H).

**Theorem 5.** *Suppose that Assumption 1 holds, and that constants $c_2$ and $c_3$ exist such that*

$$\mathbb{E}[\|g^{(k)}\|_{\sharp,*}^2] \leq c_2^2 + c_3^2\mathbb{E}[\langle\nabla f_\Phi(\nabla\Phi(x^{(k-1)})), \mathbb{E}[g^{(k)}|\mathcal{F}^{(k-1)}]\rangle]. \tag{6}$$

*In addition, suppose that the step size $\eta_k$ in Algorithm 1 satisfies $\eta_k = \Theta(1/\sqrt{k})$ and $\eta_k \leq 2\mu c_3^{-2}M^{-1}$ for all $k$, and a constant $c_4$ exists such that $f(x^{(0)}) - f^* \leq c_4$. Then,*

$$\min_{0\leq i\leq k}\mathbb{E}[\langle\nabla f_\Phi(\nabla\Phi(x^{(i)})), \mathbb{E}[g^{(i)}|\mathcal{F}^{(i-1)}]\rangle] = \mathcal{O}(\log k/\sqrt{k}).$$

Note that if (5) holds with $\lambda_k \equiv c_2^2$ and $\rho = c_3^2$, then (6) holds by taking expectation on both sides of (5). Theorem 5 states the rate at which the inner product between the pseudo-gradient and the actual gradient vanishes under just the smoothness assumption. Faster rates and global convergence can be achieved under stronger assumptions. Below we introduce the convergence rate when the objective satisfies a generalized version of the well-known Polyak-Łojasiewicz condition Polyak [1963].

**Global convergence under Polyak-Łojasiewicz condition.** We introduce our assumption below.

**Assumption 2** (Generalized Polyak-Łojasiewicz condition)**.** *For any $x \in \text{int}(\mathcal{H}_+)$, suppose*

$$\frac{1}{2}\|\nabla f_\Phi(\nabla\Phi(x))\|^2 \geq \gamma(f(x) - f^*) \quad \text{for some universal constant } \gamma > 0.$$

The above assumption generalizes the Polyak-Łojasiewicz condition [Polyak, 1963], which corresponds to the specific choice of $\Phi(x) = \|x\|^2/2$. Under this choice of $\Phi$, pseudo mirror descent reduces to pseudo gradient descent, and converges linearly [Poljak and Tsypkin, 1973]. Note that Assumption 2 is a slightly more restrictive condition than the Polyak-Łojasiewicz condition because, by chain rule, it implies the Polyak-Łojasiewicz condition so long as $\nabla^2\Phi(x)$ has bounded eigenvalues.

**Theorem 6.** *Suppose Assumptions 1, 2 and Equation* (6) *hold, and a constant $c_1 > 0$ exists such that, for all $x^{(k)}$ satisfying $f(x^{(k)}) \neq f^*$,*

$$\mathbb{E}[\langle \nabla f_\Phi(\nabla \Phi(x^{(k-1)})), \mathbb{E}[g^{(k)}|\mathcal{F}^{(k-1)}]\rangle] \geq c_1 \mathbb{E}\|\nabla f_\Phi(\nabla \Phi(x^{(k)}))\|^2, \quad \forall k \geq 1. \qquad (7)$$

*If we set $\eta_k \equiv \eta < \min\{1/(2\gamma c_1), 2M^{-1}\mu c_3^{-2}\}$, then*

$$\mathbb{E}[f(x^{(k)}) - f^*] \leq \left(1 - 2\gamma c_1 \left(\eta - \frac{M\mu^{-1}\eta^2}{2}c_3^2\right)\right)^k [f(x^{(0)}) - f^*] + \frac{M\mu^{-1}\eta^2}{2}c_2^2.$$

*If instead we set $\eta_k = \min\{(2k+1)/[\gamma c_1(k+1)^2], M^{-1}\mu c_3^{-2}\}$, then*

$$\mathbb{E}[f(x^{(k)}) - f^*] \leq \frac{M\mu^{-1}c_2^2}{2\gamma^2 c_1^2 k} \quad for \ k \geq Mc_3^2/(\gamma c_1 \mu).$$

The proof of Theorem 6 is given in Appendix I, and is built on Karimi et al. [2016], in which the same rate is obtained for stochastic gradient descent under standard Polyak-Łojasiewicz condition in an Euclidean space. By comparison, Theorem 6 is a more general result: (i) it applies to stochastic mirror descent on Hilbert spaces, (ii) it applies to any pseudo-gradient satisfying (7). As it turns out, the flexibility in utilizing pseudo-gradients instead of unbiased stochastic gradients plays an important role in many practical applications, as we will illustrate in the following section.

## 3 Pseudo Mirror Descent for Point Process Estimation

In this section, we apply pseudo mirror descent to the problems of learning the intensity functions of Poisson processes, as well as triggering functions of multivariate Hawkes processes.

### 3.1 Learning Poisson Intensities with Pseudo Mirror Descent

For simplicity of exposition, we consider a one-dimensional Poisson process over $[0, 1]$ with intensity $x^*(t)$. The objective for estimating $x^*(t)$ is

$$f(x) = \int_0^1 x(t)\mathrm{d}t - \int_0^1 x^*(t)\log x(t)\mathrm{d}t. \qquad (8)$$

This objective can be viewed as the expectation of the negative log-likelihood of a Poisson process over infinite number of sample paths. Our goal is to minimize $f(x)$ over $x \in \mathrm{int}(\mathcal{H}_+)$ with $\mathcal{H} = \mathcal{L}_2[0, 1]$. We restrict $x$ to be continuous, and we choose the generalized $I$-divergence as the Bregman divergence, with $\Phi(x) \triangleq \langle x, \log(x) - 1\rangle$.

**Deriving pseudo-gradients.** We have $\nabla \Phi(x) = \log x$, $\nabla \Phi^*(x) = \exp(x)$, and

$$f_\Phi(y) = \int_0^1 \exp(y)(t)\mathrm{d}t - \int_0^1 x^*(t)y(t)\mathrm{d}t.$$

Hence, $\nabla f_\Phi(\nabla \Phi(x)) = x - x^*$. In practice, we cannot simply choose $\nabla f_\Phi(\nabla \Phi(x))$ as the pseudo-gradient, since $x^*(t)$ is unknown, and instead only sample arrivals from the Poisson process are observed. Hence, we choose the pseudo-gradient as

$$g(t) = \int_0^1 x(\tau)K(t,\tau)\mathrm{d}\tau - \sum_{i=1}^N K(\tau_i, t),$$

where $K(\cdot, \cdot)$ is a positive definite kernel, and $\tau_1, \ldots, \tau_N$ are arrival times from the Poisson process. The introduction of $K(\cdot, \cdot)$ is necessary to avoid the presence of Dirac's delta functions in the expression of the pseudo-gradient. Substitute $x$ with $x^{(k)}$ in the expression of $g$. The resulting $g^{(k)}$ is a pseudo-gradient since $\mathbb{E}[g^{(k)}|\mathcal{F}^{(k-1)}]$ is the kernel embedding of $\nabla f_\Phi(\nabla \Phi(x^{(k-1)}))$.

**On convergence of pseudo mirror descent.** We verify that the conditions in Theorem 6 hold. When $K(\cdot, \cdot)$ is a finite-dimensional kernel (e.g., a polynomial kernel), we have

$$\langle \mathbb{E}[g^{(k)}|\mathcal{F}^{(k-1)}], \nabla f_\Phi(\nabla \Phi(x^{(k-1)}))\rangle = \int_0^1 \int_0^1 (x^{(k-1)} - x^*)(t_1)K(t_1, t_2)(x^{(k-1)} - x^*)(t_2)\mathrm{d}t_1\mathrm{d}t_2,$$

which is lower bounded by $\lambda_{\min}\|x^{(k-1)} - x^*\|^2$ where $\lambda_{\min}$ is the minimum eigenvalue of the integral operator associated with $K(\cdot,\cdot)$. This design guarantees that (7) holds.

The expected log-likelihood objective in (8) is not particularly nice for learning positive functions: as $\|x\|_\infty$ approaches 0, $f(x)$ becomes non-smooth, and violates the generalized Polyak-Łojasiewicz condition. Nevertheless, for finite number of iterations, it is reasonable to assume that the extreme values of $x(t)$ are bounded and thus the following proposition follows.

**Proposition 7.** *Consider objective* (8) *and let* $\Phi(x) = \langle x, \log x - 1\rangle$. *Then,*

- *The $\mu$-strong-convexity of $\Phi$ and* (2) *are satisfied for the $\mathcal{L}_1$-norm when $\|x\|_{\mathcal{L}_1} \leq \mu^{-1}$.*

- *The objective satisfies Assumption 2 with constant $\nu$ when $\min_{t\in[0,1]} x(t) \geq 2\nu$.*

Although this proposition requires $\|x\|_{\mathcal{L}_1} \leq \mu^{-1}$ in order for $\Phi(x)$ to be $\mu$-strongly-convex and for $f_\Phi$ to be $M\mu^{-1}$-Lipschitz-smooth for constant $M$, a crude analysis shows that the updates are essentially of the form $x^{(k+1)}(t) = x^{(k)}(t)\exp(-\eta_k[\nabla f_\Phi(x^{(k)})](t)) = \mathcal{O}(\eta_k^{-1})$. Therefore, with i.i.d. sample paths of the Poisson process observed in practice, one can expect, using standard argument of concentration inequality (see e.g., Rosasco et al. [2010]), that such condition would hold with high probability for the constant step size specified in Theorem 6. Indeed, in the next section we show that, although Polyak-Łojasiewicz condition is not strictly satisfied, a linear convergence behavior at early stage can still be observed. Meanwhile, the proof of Proposition 7 also shows that (2) may hold when $\nabla^2\Phi$ does not have uniformly bounded eigenvalues over $\mathrm{int}(\mathcal{H}_+)$.

## 3.2 Learning Multivariate Hawkes Processes with Pseudo Mirror Descent

Herein, we apply pseudo mirror descent to learn the triggering functions of a multivariate Hawkes process. A $p$-dimensional multivariate Hawkes process is a set of stochastic processes whose intensity functions, denoted by $x_1^*,\ldots,x_p^*$, are causally dependent on the past arrivals [Hawkes, 1971]:

$$x_i^*(t) = x_{i0}^* + \sum_{j=1}^p \int_{-\infty}^t y_{ij}^*(t-\tau)\mathrm{d}N_j(\tau) \quad i \in \{1,\ldots,p\}. \tag{9}$$

Here, $x_{i0}^*$ is a given base intensity, $N_j(t)$ is the counting process of dimension $j$, and $y_{ij}^* \in \mathcal{H} := \mathcal{L}_2[0,1]$ is the triggering function that captures the mutual excitation impact from dimension $j$ to $i$. Our goal is to learn the $p \times p$ triggering functions by maximizing the expected log-likelihood, which can be carried out by optimizing $p$ separate objectives of the form [Yang et al., 2017]:

$$\min_{y_{i1},\ldots,y_{ip}\in\mathcal{H}} f_i(y_{i1},\ldots,y_{ip}) = \mathbb{E}\left[\int_0^T x_i(t) - x_i^*(t)\log x_i(t)\mathrm{d}t\right], \tag{10}$$

where $x_1,\ldots,x_p$ are calculated by

$$x_i(t) = x_{i0}^* + \sum_{j=1}^p \int_{-\infty}^t y_{ij}(t-\tau)\mathrm{d}N_j(\tau) \quad i \in \{1,\ldots,p\}. \tag{11}$$

**Deriving pseudo-gradients.** We consider $\Phi(x) = \langle x, \log x - 1\rangle$. After some calculations, the partial derivative of $f_\Phi$ with respect to $\nabla\Phi(y_{ij})$ can be expressed as:

$$[\partial_{\nabla\Phi(y_{ij})}f_\Phi(\nabla\Phi(y_{i1}),\ldots,\nabla\Phi(y_{ip}))](s) = \mathbb{E}\left[\int_0^T\left(1 - \frac{x_i^*(t)}{x_i(t)}\right)y_{ij}(s)x_j^*(t-s)\mathrm{d}t\right],$$

where $s > 0$ (due to causality), and the expectation is over the sample paths. We choose the pseudo-gradient to be the kernel embedding of the above and $x^*(t)$ are accessed through samples:

$$g_{ij}(s) = \int_0^T \sum_{k=1}^{N_j(t)} K(s, t-t_{jk})y_{ij}(t-t_{jk})\mathrm{d}t - \sum_{m=0}^{N_i(T)}\sum_{n=0}^{N_j(t_{im})} \frac{K(s, t_{im}-t_{jn})}{x_i(t_{im})}y_{ij}(t_{im}-t_{jn}), \tag{12}$$

where $t_{im}$ is the $m$-th arrival in the $i$-th dimension (see Appendix for detailed construction).

**Remark 8** (On efficient representation of the updates). *For both Poisson and multivariate Hawkes processes, the updates can be tracked pointwise. If we replace the integration over $y_{ij}$ or $x$ by sample averages, $g(s)$ and $g_i(s)$ become linear combination of the kernels. This allows us to perform updates by merely keeping track of the coefficients and the parameters of those kernels.*

# 4 Numerical Experiment

In this section, we present numerical results on synthetic and real datasets. With synthetic data, the goal is to verify the results of Theorem 6, and to compare the performance of pseudo mirror descent with the link function and projection approaches mentioned in the introduction. Meanwhile, the experiment on real data is designed to show the practical performance of pseudo mirror descent. We conduct experiment with various choices of kernels, including the polynomial kernel $K(x, y) = (1 + xy)^2$, and the Sobolev kernel $K(x, y) = 1 + \min\{x, y\}$. As noted in Theorem 6, a finite-dimensional kernel guarantees (7), whereas an infinite-dimensional kernel has a better representation capability, and hence a better performance when fewer iterations are performed. Detailed parameter settings and additional results can be found in Appendices K and L.

**Learning a synthetic one-dimensional Poisson process.** We set $x^*(t) = \exp(-t)$, and evaluated the performance of pseudo mirror descent under constant and vanishing step sizes. The result is shown in Figure 1, where we plotted $\log(f(x^{(k)}) - f^*)$ versus $k$ under constant (left) and vanishing step sizes (mid), and compared the estimation errors between pseudo mirror descent, projected gradient descent, and the link function approach (right). The pseudo-gradient is calculated using a mini-batch of 10 realizations and a polynomial kernel $K(x, y) = (1 + xy)^2$. All hyperparameters are fine-tuned and reported in Appendix K. From the left-most subplot, we see that, even though the objective does not satisfy the Polyak-Łojasiewicz condition, we still observe linear convergence under a constant step size at the initial stages. From the right-most subplot, we see that the pseudo mirror descent achieves a faster convergence comparing to the link function approach and projected gradient descent. An extension of this experiment is carried out in Figure 2, where the underlying intensity function is set to a discontinuous function $x^*(t) = 1 + \lfloor 10t \rfloor$ for $t \in [0, 1]$. The left-hand side of Figure 2 shows that both Sobolev and polynomial kernel can learn a continuous approximation of the intensity function. The right-hand side of the figure shows that the Sobolev kernel has a slightly better representation power and thus a slightly better overall performance in the given number of iterations.

**Learning shot distances in professional basketball games.** We used the shot distance data of several professional basketball players over 500 games (available at `stats.nba.com`). We applied pseudo mirror descent, the link function approach, and a neural network estimator built with PyTorch [Paszke et al., 2017] to learn each player's shooting distance modeled as a Poisson process. The pseudo-gradient is computed with a Sobolev kernel $K(x, y) = 1 + \min\{x, y\}$ [Wahba, 1990], and the hyperparameters are fine-tuned and reported in Appendix K. Figure 3 depicts the result with the histogram of the data in background. We can see that the pseudo mirror descent shows a similar accuracy compared to the link function approach and the neural network estimator.

**Online learning for multivariate Hawkes process.** We studied the mouse embryonic stem cell data, which is often modeled as a multivariate Hawkes process. The dataset we adopted [Chen et al., 2008] consists of 15 DNA sequences, where each sequence documents the co-occurrence of 15 types of transcriptional regulatory elements (TREs). We modeled each DNA sequence as a 15-dimensional Hawkes process, following the setting of [Carstensen et al., 2010]. Our goal is to compare the log-likelihood per dimension, (10), evaluated using the estimates of pseudo mirror descent, the expectation maximization (EM) algorithm [Lewis and Mohler, 2011], and the MLE-SGLP proposed by Xu et al. [2016]. The pseudo-gradient is computed with the Sobolev kernel introduced above.

Figure 4 shows two scatter plots of performance comparison, between pseudo mirror descent and the EM algorithm (left), and between pseudo mirror descent and MLE-SGLPL (right). The horizontal axis is the log-likelihood of the benchmarks, implemented with Bacry et al. [2017], and the vertical axis is the log-likelihood of the pseudo mirror descent. As each dot represents the per-dimensional log-likelihood of one TRE in one DNA sequence, there are a total of $15 \times 15 = 225$ dots. We can see that, on the left-hand subplot in Figure 4, most dots fall to the left of the diagonal line, indicating that pseudo mirror descent is slightly better than the EM algorithm; on the right-hand subplot, most dots fall in the vicinity of the diagonal line, implying similar performances between pseudo mirror descent and MLE-SGLP. Note that both the EM algorithm and MLE-SGLP are batch learning algorithms.

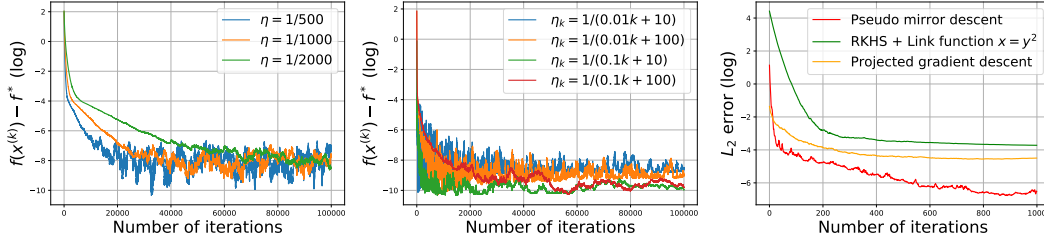

Figure 1: Synthetic dataset: log of objective error for pseudo mirror descent under constant (left) and vanishing (mid) step sizes; estimation error of pseudo mirror descent, projected gradient descent, and the link function approach (right).

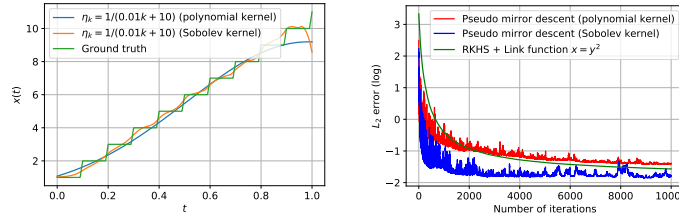

Figure 2: Synthetic dataset: the fitting of a Poisson process with piecewise constant intensity function. We compare the performance using a polynomial kernel and a Sobolev kernel (left), and compare the estimation error with the link function approach (right).

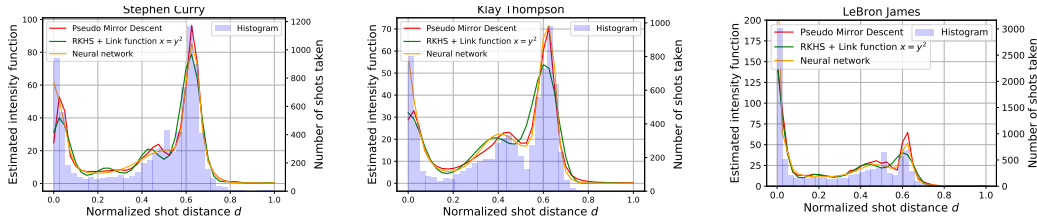

Figure 3: Basketball shot distance dataset: recovery of the intensities using pseudo mirror descent (red curve), the link function approach), and neural networks (yellow curve).

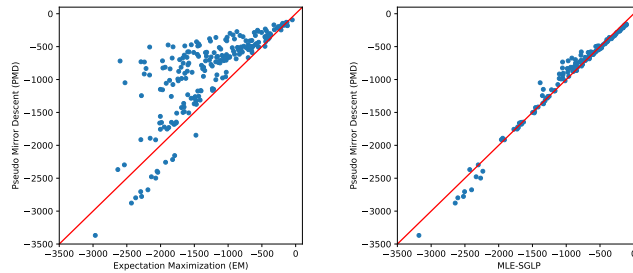

Figure 4: Mouse embryonic stem cell dataset: scatter plot comparison between pseudo mirror descent and expectation maximization (left), and between pseudo mirror descent and MLE-SGLP (right).

## 5    Conclusion

This paper introduced a principle algorithm, pseudo mirror descent, and a new theoretical framework for nonparametric estimation of positive functions. Convergence results on pseudo mirror descent apply to general-purpose (non-convex) optimization problems, which can be of independent interest. We provided examples on applying pseudo mirror descent to learning intensity and triggering functions of Poisson and multivariate Hawkes processes. Besides its strong theoretical guarantees, numerical results also showed that pseudo mirror descent generates near optimal performance in practice.

## Footnotes

[2]See Appendix B for details. The two norms $\|\cdot\|$ and $\|\cdot\|_\sharp$ need not be the same.

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
