[Supplementary Material]

# Appendix

## Table of Contents

## A  Illustration of Pseudo Mirror Descent Update

Figure 5: Illustration of pseudo mirror descent update procedure.

Figure 5 illustrates the update procedure of the pseudo mirror descent algorithm. The black solid lines in the primal space represent the update trajectory of pseudo mirror descent. The corresponding update trajectory in the dual space is represented by the black dashed lines. The red lines represent the update trajectory using the exact gradient. The angle between the pseudo-gradient and the exact gradient, denoted by $\alpha$, is acute. Intuitively, this guarantees that the pseudo mirror descent has an update trajectory in the dual space that is close to the one using the exact gradient.

## B  Duality between Strong Convexity and Lipschitz Smoothness

**Lemma 9.** *Let $\|\cdot\|_\sharp$ and $\|\cdot\|_{\sharp,*}$ be a pair of norm and dual norm. If $\Phi$ is $\mu$-strongly-convex with respect to $\|\cdot\|_\sharp$, then $\Phi^*$, the Fenchel conjugate of $f$, is $\mu^{-1}$-Lipschitz-smooth with respect to $\|\cdot\|_{\sharp,*}$.*

*Proof.* Consider $\Phi$ to be $\mu$-strongly-convex with respect to norm $\|\cdot\|_\sharp$. Then,

$$\Phi(x) \geq \Phi(y) + \langle \nabla\Phi(y), x - y \rangle + \frac{\mu}{2}\|x - y\|_\sharp^2. \tag{13}$$

We wish to prove

$$\Phi^*(x) \leq \Phi^*(y) + \langle \nabla\Phi^*(y), x - y \rangle + \frac{\mu^{-1}}{2}\|x - y\|_{\sharp,*}^2 \tag{14}$$

for any $x, y \in \mathcal{H}$. Since $\nabla\Phi(x)$ is defined through an isometric isomorphism between $\mathcal{H}$ and $\mathcal{H}^*$, it suffices to prove

$$\Phi^*(x) \leq \Phi^*(\nabla\Phi(y)) + \langle \nabla\Phi^*(\nabla\Phi(y)), x - \nabla\Phi(y) \rangle + \frac{\mu^{-1}}{2}\|x - \nabla\Phi(y)\|_{\sharp,*}^2, \tag{15}$$

or equivalently

$$\Phi^*(x) \leq \Phi^*(\nabla\Phi(y)) + \langle y, x - \nabla\Phi(y) \rangle + \frac{\mu^{-1}}{2}\|x - \nabla\Phi(y)\|_{\sharp,*}^2. \tag{16}$$

The proof exploits the following simple idea. Let $\Phi$ and $\Psi$ be mappings from $\mathcal{H}$ to $[-\infty, \infty]$, and let $\Phi(x) \leq \Psi(x)$ for all $x \in \mathcal{H}$. Then, by the definition of Fenchel conjugate,

$$\Psi^*(y) = \sup_{x \in \mathcal{H}}\{\langle x, y \rangle - \Psi(x)\} = \langle x^*(y), y \rangle - \Psi(x^*(y))$$

$$\leq \langle x^*(y), y \rangle - \Phi(x^*(y)) \leq \sup_{x \in \mathcal{H}}\{\langle x, y \rangle - \Phi(x)\} = \Phi^*(y),$$

where we denote $x^*(y)$ as the solution to $\sup_{x \in \mathcal{H}}\{\langle x, y \rangle - \Psi(x)\}$. We shall show that, if, for any fixed $y \in \mathcal{H}$, let the right-hand side of (13) be $\Psi(x)$, then the right-hand side of (16) is the expression of $\Psi^*(x)$.

To show this, let

$$\Psi(x) = \Phi(y) + \langle \nabla\Phi(y), x - y \rangle + \frac{\mu}{2}\|x - y\|_\sharp^2.$$

Since $\Phi$ is strongly convex, we have $\Phi(x) \geq \Psi(x)$, and $\Phi^*(x) \leq \Psi^*(x)$. However,

$$
\begin{aligned}
\Psi^*(x) &= \sup_{z \in \mathcal{H}}\{\langle x, z \rangle - \Psi(z)\} \\
&= \sup_{z \in \mathcal{H}}\{\langle x, z \rangle - \Phi(y) - \langle \nabla\Phi(y), z - y \rangle - \frac{\mu}{2}\|z - y\|_\sharp^2\} \\
&= \sup_{z \in \mathcal{H}}\{-\Phi(y) + \langle x, y \rangle + \langle x - \nabla\Phi(y), z - y \rangle - \frac{\mu}{2}\|z - y\|_\sharp^2\} \\
&\leq -\Phi(y) + \langle x, y \rangle + \frac{\|x - \nabla\Phi(y)\|_{\sharp,*}^2}{2\mu},
\end{aligned}
$$

where the last step follows by invoking Cauchy-Schwarz inequality $\langle x - \nabla\Phi(y), z - y \rangle \leq \|x - \nabla\Phi(y)\|_{\sharp,*}\|z - y\|_\sharp$, and computing the extremum of a quadratic form. Finally, notice that

$$-\Phi(y) + \langle x, y \rangle = \Phi^*(\nabla\Phi(y)) + \langle y, x - \nabla\Phi(y) \rangle,$$

we have that

$$\Psi^*(x) \leq \Phi^*(\nabla\Phi(y)) + \langle y, x - \nabla\Phi(y) \rangle + \frac{\mu^{-1}}{2}\|x - \nabla\Phi(y)\|_{\sharp,*}^2.$$

Hence, we have (16), and thus (14) and eventually (13). □

**Remark 10.** *Note that the conclusion of Lemma 9 can be generalized when the strong convexity of $\Phi$ is constrained to a set $\mathcal{F} \subset \mathcal{H}$. Indeed, by examining the equivalence condition between (15) and (16), the Lipschitz smoothness holds for $x, y \in \mathrm{Span}(\nabla\Phi(\mathrm{z}), \mathrm{z} \in \mathcal{F})$. For all intents and purposes, $\mathcal{F}$ is the "primal space", as illustrated in Figure 5. It immediately follows that, as long as $x$ and $y$ belong to the "dual space", then the Fenchel conjugate of $\Phi$ is Lipschitz smooth.*

## C    On Designing Pseudo-Gradient for Multivariate Hawkes Process

We consider $\Phi(x) = \langle x, \log x - 1 \rangle$. Using chain rule, the partial derivative of $f_\Phi$ with respect to $\nabla\Phi(y_{ij})$ can be expressed as:

$$[\partial_{\nabla\Phi(y_{ij})} f_\Phi(\nabla\Phi(y_{i1}), \ldots, \nabla\Phi(y_{ip}))](s) = \mathbb{E}\left[\int_0^T \left(1 - \frac{x_i^*(t)}{x_i(t)}\right) y_{ij}(s) x_j^*(t - s)\mathrm{d}t\right],$$

where $s > 0$ (due to causality), and the expectation is over the sample paths. We choose the pseudo-gradient to be the kernel embedding of the above and $x^*(t)$:

$$g_{ij}(s) = \int_0^T K(s, r)\mathrm{d}r \int_0^T \left(1 - \frac{x_i^*(t)}{x_i(t)}\right) y_{ij}(r) x_j^*(t - r)\mathrm{d}t.$$

In practice, the value of $x^*(t)$ can only be accessed through samples. Hence, we choose

$$g_{ij}(s) = \int_0^T \sum_{k=1}^{N_j(t)} K(s, t - t_{jk}) y_{ij}(t - t_{jk})\mathrm{d}t - \sum_{m=0}^{N_i(T)} \sum_{n=0}^{N_j(t_{im})} \frac{K(s, t_{im} - t_{jn})}{x_i(t_{im})} y_{ij}(t_{im} - t_{jn}),$$

where $t_{im}$ is the $m$-th arrival in the $i$-th dimension. This is obtained upon approximating the integral over $x_j^*(t - r)$ and the double integral over $x_j^*(t - r)x_i^*(t)$ by their corresponding sample averages under the multivariate Hawkes process, respectively.

# D  Proof of Example 1

We observe that $f(x) = f_\Phi(\nabla\Phi(x))$. This can be proved by showing $\nabla\Phi^* = (\nabla\Phi)^{-1}$. To show this, we first suppose $y = \nabla\Phi(x)$. Then, $\Phi(x) + \Phi^*(y) = \langle x, y \rangle$. Since $\Phi^{**} = \Phi$, we have $\Phi^*(y) + \Phi^{**}(x) = \langle x, y \rangle$, which yields $x = \nabla\Phi^*(y)$. Hence, $x = \nabla\Phi^*(y) = \nabla\Phi^*(\nabla\Phi(x))$, showing $\nabla\Phi^* = (\nabla\Phi)^{-1}$.

With this observation, we further invoke the definition of directional derivative, which gives us, for $y \in \mathcal{H}$,

$$
\begin{aligned}
\langle \nabla f(x), y \rangle &= \lim_{\epsilon \to 0} \frac{f_\Phi(\nabla\Phi(x + \epsilon y)) - f_\Phi(\nabla\Phi(x))}{\epsilon} \\
&= \lim_{\epsilon \to 0} \frac{f_\Phi(\nabla\Phi(x) + \langle \nabla^2\Phi(x), \epsilon y \rangle + o(\epsilon)) - f_\Phi(\nabla\Phi(x))}{\epsilon} \\
&= \lim_{\epsilon \to 0} \frac{\langle \nabla f_\Phi(\nabla\Phi(x)), \langle \nabla^2\Phi(x), \epsilon y \rangle \rangle}{\epsilon},
\end{aligned}
$$

where the last step holds due to the Lipschitzness of $f_\Phi$, which follows from the smoothness of $f$ and the strong convexity of $\Phi$. Hence, substituting $y = \nabla f_\Phi(\nabla\Phi(x))$ and let $x = x^{(k-1)}$ gives the result.

# E  Proof of Lemma 2

The first order condition of (4) when optimizing over $\mathcal{H}$ is

$$
\nabla\Phi(x) = \nabla\Phi(x^{(k)}) - \eta_k g^{(k+1)}.
$$

As shown in Appendix D, $\nabla\Phi^* = (\nabla\Phi)^{-1}$. Hence,

$$
x = \nabla\Phi^*(\nabla\Phi(x^{(k)}) - \eta_k g^{(k+1)}).
$$

By assumption, $x \in \mathcal{H}_+$. Therefore $x = x^{(k+1)}$.

# F  Proof of Theorem 3

## F.1  A Technical Lemma

Before we state the formal part of the proof, we first prove a technical lemma, which will be used at the beginning of the main proof.

**Lemma 11.** *Suppose $f_\Phi$ is continuously differentiable, and satisfies $\|\nabla f_\Phi(x + y) - \nabla f_\Phi(x)\|_\sharp \leq L\|y\|_{\sharp,*}$. Then,*

$$
f_\Phi(x + y) - f_\Phi(x) - \langle \nabla f_\Phi(x), y \rangle \leq \frac{L}{2}\|y\|_{\sharp,*}^2.
$$

**Remark 12.** *This lemma states two equivalent ways of expressing the smoothness of $f_\Phi$. The result is standard when $\|\cdot\|_\sharp$ is the norm of the Hilbert space, which, not surprisingly, corresponds to the case where $\Phi(x) = \|x\|^2/2$. Here, we extend the proof of Theorem 18.13 in Bauschke and Combettes [2011] to show equivalence for more general choices of $\|\cdot\|_\sharp$.*

*Proof.* We start from the given condition

$$
\|\nabla f_\Phi(x + y) - \nabla f_\Phi(x)\|_\sharp \leq L\|y\|_{\sharp,*},
$$

and multiply the two sides by $\|y\|_{\sharp,*}$:

$$
\|\nabla f_\Phi(x + y) - \nabla f_\Phi(x)\|_\sharp \|y\|_{\sharp,*} \leq L\|y\|_{\sharp,*}^2.
$$

By Cauchy-Schwarz inequality, the left-hand side can be further lower bounded by $\langle \nabla f_\Phi(x + y) - \nabla f_\Phi(x), y \rangle$, and hence

$$
\langle \nabla f_\Phi(x + y) - \nabla f_\Phi(x), y \rangle \leq L\|y\|_{\sharp,*}^2. \tag{17}
$$

Now, define $h(t) = f_\Phi(x + ty)$. Then, the derivative of $h$ is $h'(t) = \langle y, \nabla f_\Phi(x + ty) \rangle$. Hence, by the fundamental theorem of calculus, we have

$$
\begin{aligned}
f_\Phi(x + y) - f_\Phi(x) - \langle \nabla f_\Phi(x), y \rangle &= \int_0^1 \langle y, \nabla f_\Phi(x + ty) - \nabla f_\Phi(x) \rangle \mathrm{d}t \\
&= \int_0^1 \frac{1}{t} \langle ty, \nabla f_\Phi(x + ty) - \nabla f_\Phi(x) \rangle \mathrm{d}t \\
&\leq \int_0^1 \frac{1}{t} L \|ty\|_{\sharp,*}^2 \mathrm{d}t \\
&= \int_0^1 Lt \|y\|_{\sharp,*}^2 \mathrm{d}t \\
&= \frac{L}{2} \|y\|_{\sharp,*}^2,
\end{aligned}
$$

where the inequality holds by applying (17).

$\square$

## F.2 The Main Proof

The proof follows the procedure of Poljak and Tsypkin [1973]. The key to proving the existence of $\lim_{k \to \infty} f(x^{(k)})$ is to show that $f(x^{(k)}) - f^*$ is a semimartingale, and that its limit exists almost surely. The key to proving the asymptotic convergence of $\langle \nabla f_\Phi(\nabla \Phi(x^{(k)})), \mathbb{E}[g^{(k)}|\mathcal{F}^{(k-1)}] \rangle$ is to show $\lim_{k \to \infty} \eta_k \mathbb{E}[\langle \nabla f_\Phi(\nabla \Phi(x^{(k)})), \mathbb{E}[g^{(k)}|\mathcal{F}^{(k-1)}] \rangle] = 0$, which then yields the result upon noticing that $\langle \nabla f_\Phi(\nabla \Phi(x^{(k)})), \mathbb{E}[g^{(k)}|\mathcal{F}^{(k-1)}] \rangle \geq 0$ by design. Below are the details of the proof.

First observe that for a pseudo mirror descent update $x^{(k)}$, we have $f(x^{(k)}) = f_\Phi(\nabla \Phi(x^{(k)}))$ according to the proof in Appendix D. Incorporating this fact with Lemma 11, and setting $L = M\mu^{-1}$, we immediately have

$$
\begin{aligned}
f(x^{(k)}) - f(x^{(k-1)}) - \langle \nabla f_\Phi(x^{(k-1)}), \nabla \Phi(x^{(k)}) - \nabla \Phi(x^{(k-1)}) \rangle & \\
&\leq \frac{M\mu^{-1}}{2} \|\nabla \Phi(x^{(k)}) - \nabla \Phi(x^{(k-1)})\|_{\sharp,*}^2
\end{aligned}
$$

for pseudo mirror descent updates. By Lemma 2, the pseudo mirror descent updates also satisfy $\nabla \Phi(x^{(k)}) - \nabla \Phi(x^{(k-1)}) = -\eta_{k-1} g^{(k)}$. Therefore,

$$
f(x^{(k)}) - f(x^{(k-1)}) + \langle \nabla f_\Phi(\nabla \Phi(x^{(k-1)})), \eta_{k-1} g^{(k)} \rangle \leq \frac{M\mu^{-1}\eta_{k-1}^2}{2} \|g^{(k)}\|_{\sharp,*}^2,
$$

which further implies

$$
f(x^{(k)}) \leq f(x^{(k-1)}) - \eta_{k-1} \langle \nabla f_\Phi(\nabla \Phi(x^{(k-1)})), g^{(k)} \rangle + \frac{M\mu^{-1}\eta_{k-1}^2}{2} \|g^{(k)}\|_{\sharp,*}^2. \tag{18}
$$

Taking conditional expectations on both sides of (18),

$$
\begin{aligned}
\mathbb{E}[f(x^{(k)}) - f^*|\mathcal{F}^{(k-1)}] &\leq (f^{(k-1)} - f^*) - \eta_{k-1} \langle \nabla f_\Phi(\nabla \Phi(x^{(k-1)})), \mathbb{E}[g^{(k)}|\mathcal{F}^{(k-1)}] \rangle + \\
&\quad + \frac{M\mu^{-1}\eta_{k-1}^2}{2} \mathbb{E}[\|g^{(k)}\|_{\sharp,*}^2|\mathcal{F}^{(k-1)}] \\
&\leq -\eta_{k-1} \langle \nabla f_\Phi(\nabla \Phi(x^{(k-1)})), \mathbb{E}[g^{(k)}|\mathcal{F}^{(k-1)}] \rangle \left(1 - \frac{M\mu^{-1}}{2}\eta_{k-1}\right) + \\
&\quad + \frac{M\mu^{-1}\lambda_k \eta_{k-1}^2}{2} + (f^{(k-1)} - f^*) \\
&\leq (f^{(k-1)} - f^*) + \frac{M\mu^{-1}\lambda_k \eta_{k-1}^2}{2}, \tag{19}
\end{aligned}
$$

where the last step holds for sufficiently large $k$. Let

$$z^{(k)} = (f(x^{(k)}) - f^*) + \sum_{i \geq k} \frac{M\mu^{-1}\lambda_i \eta_{i-1}^2}{2}.$$

Then, we immediately have, upon substituting $z^{(k)}$ into (19),

$$\mathbb{E}[z^{(k)}|\mathcal{F}^{(k-1)}] \leq z^{(k-1)}.$$

Since for any sequence $z^{(k)}$, $f(x^{(k)}) - f^*$ can be uniquely determined (nothing else is random), we can take conditional expectations on both sides of (19) with respect to $z^{(1)}, \ldots, z^{(k-1)}$, and obtain

$$\mathbb{E}[z^{(k)}|z^{(1)}, \ldots, z^{(k-1)}] \leq z^{(k-1)}.$$

This shows that $z^{(k)}$ is a semimartingale, and that $\mathbb{E}z^{(k)} \leq \cdots \leq \mathbb{E}z^{(1)} < \infty$. This implies $\lim_{k\to\infty} z^{(k)}$ exists almost surely, and hence, $\lim_{k\to\infty}(f(x^{(k)}) - f^*)$ exists almost surely, and that $\mathbb{E}[f(x^{(k)}) - f^*]$ are uniformly upper bounded.

Taking unconditional expectations on both sides of (19), we now have

$$\mathbb{E}[f(x^{(k)}) - f^*] \leq \mathbb{E}[f(x^{(k-1)}) - f^*] + \frac{M\mu^{-1}\lambda_k \eta_{k-1}^2}{2} -$$
$$- \eta_{k-1}\mathbb{E}[\langle \nabla f_\Phi(\nabla\Phi(x^{(k-1)})), \mathbb{E}[g^{(k)}|\mathcal{F}^{(k-1)}]\rangle]\left(1 - \frac{M\mu^{-1}\rho}{2}\eta_{k-1}\right).$$

For sufficiently large $k$, $2 - M\mu^{-1}\rho\eta_k > 0$. Hence, summing both sides from $k = 1$ to $\infty$, we get

$$\mathbb{E}[f(x^{(0)}) - f^*] + \sum_{k=0}^{\infty} \frac{M\mu^{-1}\lambda_{k+1}\eta_k^2}{2} \geq \sum_{k=0}^{\infty} \eta_k \mathbb{E}[\langle \nabla f_\Phi(\nabla\Phi(x^{(k)})), \mathbb{E}[g^{(k+1)}|\mathcal{F}^{(k)}]\rangle]\left(1 - \frac{M\mu^{-1}\rho}{2}\eta_k\right).$$

By assumption, the left-hand side of the above inequality is finite. In other words,

$$\sum_{k=0}^{\infty} \eta_k \mathbb{E}[\langle \nabla f_\Phi(\nabla\Phi(x^{(k)})), \mathbb{E}[g^{(k+1)}|\mathcal{F}^{(k)}]\rangle]\left(1 - \frac{M\mu^{-1}\rho}{2}\eta_k\right) < \infty.$$

On the other side, we also have $\sum_{k=0}^{\infty} \eta_k = \infty$, $\langle \nabla f_\Phi(\nabla\Phi(x^{(k)})), \mathbb{E}[g^{(k+1)}|\mathcal{F}^{(k)}]\rangle \geq 0$, while for sufficiently large $k$, $1 - \eta_k M\mu^{-1}\rho/2 \geq \varepsilon > 0$ for some small constant $\varepsilon$. Therefore, there exists a subsequence $k_i$ such that $\langle \nabla f_\Phi(\nabla\Phi(x^{(k)})), \mathbb{E}[g^{(k+1)}|\mathcal{F}^{(k)}]\rangle$ converges in distribution:

$$\lim_{i\to\infty} \mathbb{E}[\langle \nabla f_\Phi(\nabla\Phi(x^{(k_i-1)})), \mathbb{E}[g^{(k_i)}|\mathcal{F}^{(k_i-1)}]\rangle] = 0.$$

Since the sequence converges in distribution to a constant, it also converges in probability, which further implies almost sure convergence of a subsequence:

$$\lim_{j\to\infty} \langle \nabla f_\Phi(\nabla\Phi(x^{(k_{i_j}-1)})), \mathbb{E}[g^{(k_{i_j})}|\mathcal{F}^{(k_{i_j}-1)}]\rangle = 0$$

almost surely. This implies the final result.

## G   Proof of Corollary 4

**Proof when $g^{(k)} = \nabla f_\Phi(\nabla\Phi(x^{(k-1)}))$.** By the last step in proof of Theorem 3, we have

$$\lim_{k\to\infty} \|\nabla f_\Phi(\nabla\Phi(x^{(k)}))\|^2 = 0$$

in probability. By chain rule and boundedness of eigenvalues of $\nabla^2\Phi$, this further implies

$$\lim_{k\to\infty} \|\nabla f(x^{(k)})\|^2 = 0$$

in probability.

**Proof when $g^{(k)} = \nabla f(x^{(k-1)})$.** By chain rule,

$$\lim_{k\to\infty} \langle \mathbb{E}[g^{(k+1)}|\mathcal{F}^{(k)}], \nabla f_\Phi(\nabla\Phi(x^{(k)}))\rangle = \lim_{k\to\infty} \langle \nabla f_\Phi(\nabla\Phi(x^{(k)})), [\nabla^2\Phi(x^{(k)})](\nabla f_\Phi(\nabla\Phi(x^{(k)})))\rangle.$$

By Theorem 3, the left-hand side converges to 0 in probability, while the right-hand side is lower bounded by $\lim_{k\to\infty} \lambda_{\min}\|\nabla f_\Phi(\nabla\Phi(x^{(k)}))\|^2$. Therefore,

$$\lim_{k\to\infty} \|\nabla f_\Phi(\nabla\Phi(x^{(k)}))\| = 0$$

in probability. Since $\lambda_{\max}$ is upper bounded, we also have

$$\lim_{k\to\infty} \|\nabla f(x^{(k)})\| = 0$$

in probability by chain rule.

## H   Proof of Theorem 5

First, we take unconditional expectation on both sides of (19), which gives

$$
\begin{aligned}
\mathbb{E}[f(x^{(k)}) - f^*] &\leq \mathbb{E}[f(x^{(k-1)}) - f^*] - \eta_{k-1}\mathbb{E}[\langle \nabla f_\Phi(\nabla\Phi(x^{(k-1)})), \mathbb{E}[g^{(k)}|\mathcal{F}^{(k-1)}]\rangle]+ \\
&\quad + \frac{M\mu^{-1}\eta_{k-1}^2}{2}\mathbb{E}[\|g^{(k)}\|_{\sharp,*}^2] \\
&\leq -\left(\eta_{k-1} - \frac{c_3^2 M\mu^{-1}\eta_{k-1}^2}{2}\right)\mathbb{E}[\langle \nabla f_\Phi(\nabla\Phi(x^{(k-1)})), \mathbb{E}[g^{(k)}|\mathcal{F}^{(k-1)}]\rangle]+ \\
&\quad + \mathbb{E}[f(x^{(k-1)}) - f^*] + \frac{M\mu^{-1}\eta_{k-1}^2 c_2^2}{2}.
\end{aligned}
$$

Telescoping both sides over $k$, and dividing both sides by $\sum_{\kappa=0}^{k-1}\eta_\kappa$, we have

$$
\begin{aligned}
&\sum_{\kappa=1}^{k} \frac{\eta_{\kappa-1} - \frac{c_3^2 M\mu^{-1}\eta_{\kappa-1}^2}{2}}{\sum_{i=1}^{k}\left(\eta_{i-1} - \frac{c_3^2 M\mu^{-1}\eta_{i-1}^2}{2}\right)}\mathbb{E}[\langle \nabla f_\Phi(\nabla\Phi(x^{(\kappa-1)})), \mathbb{E}[g^{(\kappa)}|\mathcal{F}^{(\kappa-1)}]\rangle] \\
&\leq \frac{\mathbb{E}[f(x^{(0)}) - f(x^{(k)})] + \sum_{\kappa=1}^{k}\frac{M\mu^{-1}c_2^2}{2}\eta_{\kappa-1}^2}{\sum_{\kappa=1}^{k}\left(\eta_{\kappa-1} - \frac{c_3^2 M\mu^{-1}\eta_{\kappa-1}^2}{2}\right)}.
\end{aligned}
$$

By assumption, the denominators on both sides of the above inequality are positive, and since $\eta_k = \Theta(1/\sqrt{k})$, both denominators are dominated by $\sum_{\kappa=1}^{k}\eta_\kappa = \Theta(\sqrt{k})$ for large $k$. Since $f(x^{(0)}) - f(x^{(k)}) \leq f(x^{(0)}) - f^*$, the numerator on the right-hand side is upper bounded by $c_4 + \mathcal{O}(\log k) = \mathcal{O}(\log k)$. Meanwhile, the left-hand side is lower bounded by the minimum of $\mathbb{E}[\langle \nabla f_\Phi(\nabla\Phi(x^{(\kappa-1)})), \mathbb{E}[g^{(\kappa)}|\mathcal{F}^{(\kappa-1)}]\rangle]$ for $\kappa \in \{1,\ldots,k\}$. Hence, we reached the conclusion.

## I   Proof of Theorem 6

By Assumption 1, we know that $f_\Phi$ is $M\mu^{-1}$-smooth, and that

$$f(x^{(k)}) \leq f(x^{(k-1)}) - \eta_{k-1}\langle \nabla f_\Phi(\nabla\Phi(x^{(k-1)})), g^{(k)}\rangle + \frac{M\mu^{-1}\eta_{k-1}^2}{2}\|g^{(k)}\|_{\sharp,*}^2.$$

Taking unconditional expectations on both sides, we have

$$
\begin{aligned}
\mathbb{E}[f(x^{(k)}) - f^*] &\leq \mathbb{E}[f(x^{(k-1)}) - f^*] - \eta_{k-1}\mathbb{E}[\langle \nabla f_\Phi(\nabla\Phi(x^{(k-1)})), \mathbb{E}[g^{(k)}|\mathcal{F}^{(k-1)}]\rangle]+ \\
&\quad + \frac{M\mu^{-1}\eta_{k-1}^2}{2}\mathbb{E}[\|g^{(k)}\|_{\sharp,*}^2].
\end{aligned}
$$

By the upper bound assumption on $\mathbb{E}[\|g^{(k)}\|_{\sharp,*}^2]$ and the lower bound assumption on $\mathbb{E}[\langle \nabla f_\Phi(\nabla \Phi(x^{(k-1)})), \mathbb{E}[g^{(k)}|\mathcal{F}^{(k-1)}]\rangle]$, we further have

$$\mathbb{E}[f(x^{(k)}) - f^*] \leq \mathbb{E}[f(x^{(k-1)}) - f^*] - c_1\left(\eta_{k-1} - \frac{M\mu^{-1}\eta_{k-1}^2}{2}c_3^2\right)\mathbb{E}[\|\nabla f_\Phi(\nabla \Phi(x^{(k-1)}))\|^2] +$$
$$+ \frac{M\mu^{-1}\eta_{k-1}^2}{2}c_2^2.$$

Notice that here we have already used the assumption that $\eta_k \leq 2/(M\mu^{-1}c_3^2)$, which is required in both cases of constant and diminishing step sizes, to guarantee the positivity of the quantity $\eta_{k-1} - 0.5M\mu^{-1}\eta_{k-1}^2c_3^2$. By Assumption 2, we further have

$$\mathbb{E}[f(x^{(k)}) - f^*] \leq \mathbb{E}[f(x^{(k-1)}) - f^*] - 2\gamma c_1\left(\eta_{k-1} - \frac{M\mu^{-1}\eta_{k-1}^2}{2}c_3^2\right)\mathbb{E}[f(x^{(k-1)}) - f^*] +$$
$$+ \frac{M\mu^{-1}\eta_{k-1}^2}{2}c_2^2$$
$$= \mathbb{E}[f(x^{(k-1)}) - f^*]\left\{1 - 2\gamma c_1\left(\eta_{k-1} - \frac{M\mu^{-1}\eta_{k-1}^2}{2}c_3^2\right)\right\} + \frac{M\mu^{-1}\eta_{k-1}^2}{2}c_2^2.$$

The remaining part of the proof coincides with that of Theorem 4 in Karimi et al. [2016].

**Constant step size.** Choosing $\eta_k \equiv \eta < \min\{1/(2\gamma c_1), 2M^{-1}\mu c_3^{-2}\}$, the factor in the curled brackets is strictly between 0 and 1. Hence, we have

$$\mathbb{E}[f(x^{(k)}) - f^*] \leq \left(1 - 2\gamma c_1\left(\eta_{k-1} - \frac{M\mu^{-1}\eta_{k-1}^2}{2}c_3^2\right)\right)^k [f(x^{(0)}) - f^*] + \frac{M\mu^{-1}\eta^2}{2}c_2^2$$

upon telescoping over $k$.

**Decreasing step size.** Choosing $\eta_k = \min\{(2k+1)/[\gamma c_1(k+1)^2], M^{-1}\mu c_3^{-2}\}$, we get

$$\eta_{k-1} - \frac{M\mu^{-1}\eta_{k-1}^2}{2}c_3^3 \geq \eta_{k-1} - \eta_{k-1}\cdot\frac{M\mu^{-1}c_3^2}{2}\cdot M^{-1}\mu c_3^{-2} = \frac{\eta_{k-1}}{2}.$$

Hence,

$$\mathbb{E}[f(x^{(k+1)}) - f^*] \leq \frac{k^2}{(k+1)^2}\mathbb{E}[f(x^{(k)}) - f^*] + \frac{M\mu^{-1}c_2^2}{2}\cdot\frac{(2k+1)^2}{\gamma^2 c_1^2(k+1)^4}$$

for large $k$ such that $\eta_k = (2k+1)/[\gamma c_1(k+1)^2]$. This holds true when $k \geq Mc_3^2/(\gamma c_1\mu)$. Multiplying both sides by $(k+1)^2$, and letting $\delta(k+1) = k^2\mathbb{E}[f(x^{(k)}) - f^*]$, we get

$$\delta(k+1) \leq \delta(k) + \frac{M\mu^{-1}c_2^2}{2\gamma^2 c_1^2},$$

which holds since $(k+1)^4 \geq (2k+1)^2$. Summing from 0 to $k$ and using the fact that $\delta(0) = 0$, we get

$$(k+1)^2\mathbb{E}[f(x^{(k+1)}) - f^*] \leq \frac{M\mu^{-1}c_2^2}{2\gamma^2 c_1^2}(k+1),$$

and we reach the conclusion.

## J   Proof of Proposition 7

In this section, we verify three most important factors in the claim of Proposition 7. These include the (conditional) strong convexity of $\Phi(x)$ and the smoothness of the objective $f_\Phi$, as well as the (conditional) satisfactory of the generalized Polyak-Łojasiewicz assumption.

## J.1 Strong Convexity of $\Phi(x) = \langle x, \log x - 1 \rangle$ for $x$ with Bounded $\mathcal{L}_1$-Norm

We prove the following lemma by generalizing the proof of Pinsker's inequality by Pollard in his course notes [Pollard, 2005].

**Lemma 13** (Generalized Pinsker's inequality). *Assume that $\mathcal{H} = \mathcal{L}_2[0,1]$. Let $\Phi(x) = \langle x, \log x - 1 \rangle$. Then $\Phi(x)$ is $\mu$-strongly-convex with respect to $\mathcal{L}_1$-norm when $\|x\|_{\mathcal{L}_1} \leq \mu^{-1}$. That is, for $x$ and $y$ satisfying $\max\{\|x\|_{\mathcal{L}_1}, \|y\|_{\mathcal{L}_1}\} \leq \mu^{-1}$, we have*

$$\Delta_\Phi(x,y) \geq \frac{\mu}{2} \|x - y\|_{\mathcal{L}_1}^2.$$

A technical lemma required for the proof is stated below.

**Lemma 14.** *For $s > -1$,*

$$(1+s)\log(1+s) - s \geq \frac{1}{2} \cdot \frac{s^2}{1 + s^2/3}.$$

Recall that for $\Phi(x) = \int_0^1 x(t) \log x(t) \mathrm{d}t - \int_0^1 x(t) \mathrm{d}t$. We have

$$\Delta_\Phi(x,y) = \int_0^1 x(t) \log \frac{x(t)}{y(t)} \mathrm{d}t - \int_0^1 (x(t) - y(t)) \mathrm{d}t,$$

which holds for any Hilbert space that permits the interchange between the integration and the gradient. Let

$$r(t) = \frac{x(t)}{y(t)} - \frac{\|x\|_{\mathcal{L}_1}}{\|y\|_{\mathcal{L}_1}}.$$

Then, letting $\mathbb{E}_y[h(t)] := \int_0^1 y(t) h(t) \mathrm{d}t$, we have

$$\mathbb{E}_y[r(t)] = 0.$$

It is not hard to see, with simple manipulations, that

$$\Delta_\Phi(x,y) = \mathbb{E}_y \left[ \left( \frac{\|x\|_{\mathcal{L}_1}}{\|y\|_{\mathcal{L}_1}} + r(t) \right) \log \left( \frac{\|x\|_{\mathcal{L}_1}}{\|y\|_{\mathcal{L}_1}} + r(t) \right) - \left( \frac{\|x\|_{\mathcal{L}_1}}{\|y\|_{\mathcal{L}_1}} + r(t) - 1 \right) \right].$$

Invoking the technical Lemma 14 with

$$s = \frac{\|f\|_{\mathcal{L}_1}}{\|g\|_{\mathcal{L}_1}} + r(x) - 1,$$

we get

$$\Delta_\Phi(x,y) \geq \mathbb{E}_y \left[ \frac{1}{2} \cdot \frac{\left( \frac{\|x\|_{\mathcal{L}_1}}{\|y\|_{\mathcal{L}_1}} + r(t) - 1 \right)^2}{1 + \frac{1}{3} \cdot \left( \frac{\|x\|_{\mathcal{L}_1}}{\|y\|_{\mathcal{L}_1}} + r(t) - 1 \right)} \right].$$

It is not hard to notice that

$$\mathbb{E}_y \left[ 1 + \frac{1}{3} \cdot \left( \frac{\|x\|_{\mathcal{L}_1}}{\|y\|_{\mathcal{L}_1}} + r(t) - 1 \right) \right] = \|y\|_{\mathcal{L}_1} \cdot \left( \frac{2}{3} + \frac{1}{3} \cdot \frac{\|x\|_{\mathcal{L}_1}}{\|y\|_{\mathcal{L}_1}} \right).$$

Hence, incorporating this equation into the lower bound of $\Delta_\Phi(x,y)$ and applying Cauchy-Schwarz inequality, we get

$$\Delta_\Phi(x,y) \geq \frac{1}{2} \cdot \frac{\mathbb{E}_y^2 \left[ \left| \frac{\|x\|_{\mathcal{L}_1}}{\|y\|_{\mathcal{L}_1}} + r(t) - 1 \right| \right]}{\|y\|_{\mathcal{L}_1} \cdot \left( \frac{2}{3} + \frac{1}{3} \cdot \frac{\|x\|_{\mathcal{L}_1}}{\|y\|_{\mathcal{L}_1}} \right)}.$$

Notice that

$$\mathbb{E}_y \left[ \left| \frac{\|x\|_{\mathcal{L}_1}}{\|y\|_{\mathcal{L}_1}} + r(t) - 1 \right| \right] = \|x - y\|_{\mathcal{L}_1}^2,$$

we have

$$\Delta_\Phi(x,y) \geq \frac{C}{2} \cdot \|x-y\|_{\mathcal{L}_1}^2,$$

with

$$C = \left( \frac{2}{3} \cdot \|y\|_{\mathcal{L}_1} + \frac{1}{3} \cdot \|x\|_{\mathcal{L}_1} \right)^{-1}.$$

Since $\|x\|_{\mathcal{L}_1}$ and $\|y\|_{\mathcal{L}_1}$ are upper bounded by $\mu^{-1}$, we get $C \geq \mu$. Hence $\Phi(x)$ is $\mu$-strongly-convex with respect to $\mathcal{L}_1$ norm when $\|x\|_{\mathcal{L}_1} \leq \mu^{-1}$.

## J.2 Verification of Lipschitz Smoothness of $f_\Phi$ for $x$ with Bounded $\mathcal{L}_1$-Norm

From the derivations in Section 3, we have $\nabla f_\Phi(\nabla\Phi(x)) = x - x^*$. Since the Poisson process is on $[0,1]$, we have

$$\|\nabla f_\Phi(\nabla\Phi(y)) - \nabla f_\Phi(\nabla\Phi(x))\|_{\mathcal{L}_1} = \|x - y\|_{\mathcal{L}_1} \leq 2\mu^{-1},$$

by triangle inequality. On the other hand,

$$\|\nabla\Phi(y) - \nabla\Phi(x)\|_\infty = \|\log(y) - \log(x)\|_\infty.$$

By the definition of the $\mathcal{L}_1$-norm and the infinity norm, we can assume, without loss of generality, that $y(t) \geq x(t)$ for all $t \in [0,1]$. This assumption simplifies our subsequent presentation. We also assume $\|\log(y) - \log(x)\|_\infty$ is achieved at $t^*$. This reduces the subsequent analysis into two cases, which we present below.

**Case I:** First consider the case where $\log(y(t^*)) - \log(x(t^*)) \geq 2M^{-1}$. In this case, we have

$$M\mu^{-1}\|\log(y) - \log(x)\|_\infty = M\mu^{-1}(\log(x(t^*)) - \log(y(t^*))) \geq 2\mu^{-1} \geq \|x - y\|_{\mathcal{L}_1}.$$

**Case II:** Next consider the case where $\log(y(t^*)) - \log(x(t^*)) = \epsilon < 2M^{-1}$. In this case, the condition $\log(y(t^*)) - \log(x(t^*)) < 2M^{-1}$ and the previous assumption that $y(t) \geq x(t)$ for all $t \in [0,1]$ now together imply that $x$ and $y$ are close. This allows us to bound $\|\log(y) - \log(x)\|_\infty$ from below. The detailed analysis is below.

First, notice that for a given function $f$ such that $f_\Phi$ is Lipschitz smooth with constant $M\mu^{-1}$, it is also $M'\mu^{-1}$-Lipschitz-smooth with any $M' > M$. Hence, we can assume, without loss of generality, that $M > 2/\log 3$, which guarantees for any $\epsilon < 2M^{-1}$ we have $(\exp(\epsilon) - 1)\epsilon^{-1} < M$. This allows us to write

$$M\mu^{-1}\|\log(y) - \log(x)\|_\infty > \frac{\exp(\epsilon) - 1}{\epsilon} \cdot \mu^{-1}\|\log(y) - \log(x)\|_\infty$$
$$= (\exp(\epsilon) - 1)\mu^{-1}$$
$$\geq (\exp(\epsilon) - 1)\|x\|_{\mathcal{L}_1},$$

where the first inequality uses the assumption $M > (\exp(\epsilon) - 1)\epsilon^{-1}$, and the equality that followed uses the assumption that $\|\log(y) - \log(x)\|_\infty = \epsilon$. Finally, the assumption $\|x\|_{\mathcal{L}_1} \leq \mu^{-1}$ is used.

Now, notice that we have assumed, without loss of generality, that $y(t) \geq x(t)$ for all $t \in [0,1]$. This implies $0 < \log(y(t)) - \log(x(t)) \leq \epsilon$ for any $t \in [0,1]$, which further implies $1 < y/x \leq \exp(\epsilon)$. Hence, we further have

$$M\mu^{-1}\|\log(y) - \log(x)\|_\infty \geq (\exp(\epsilon) - 1)\|x\|_{\mathcal{L}_1}$$
$$= \int_0^1 x(t)(\exp(\epsilon) - 1)\mathrm{d}t$$
$$\geq \int_0^1 x(t)\left(\frac{y(t)}{x(t)} - 1\right)\mathrm{d}t$$
$$= \int_0^1 (y(t) - x(t))\mathrm{d}t$$
$$= \|y - x\|_{\mathcal{L}_1}.$$

Hence, we have verified the smoothness of the objective.

## J.3   Verification of the Polyak-Łojasiewicz Condition when $\min_{t\in[0,1]} x(t)$ is Bounded Away from 0

First notice that

$$\|\nabla f_\Phi(\nabla\Phi(x))\|^2 = \int_0^1 (x(t) - x^*(t))^2\, \mathrm{d}t,$$

and

$$2\nu(f(x) - f^*) = 2\nu \cdot \left[\int_0^1 (x - x^*)(t)\mathrm{d}t - \int_0^1 x^*(t)\log\frac{x}{x^*}(t)\mathrm{d}t\right].$$

We wish to prove a conclusion similar to the following:

$$\int_0^1 x^2(t)\left(1 - \frac{x^*}{x}(t)\right)^2 \mathrm{d}t \geq 2\nu\int_0^1 \left(x - x^* - x^*\log\frac{x}{x^*}\right)(t)\mathrm{d}t.$$

Notice that the left-hand side resembles the form of a $\chi^2$ divergence, whereas the right-hand side resembles the form of a Kullback-Leibler divergence. In fact, when $\min_{t\in[0,1]} x(t) \geq 2\nu$, we have

$$
\begin{aligned}
2\nu \cdot \left[\int_0^1 (x - x^*)(t)\mathrm{d}t - \int_0^1 x^*(t)\log\frac{x}{x^*}(t)\mathrm{d}t\right] &= 2\nu \cdot \left[\int_0^1 (x - x^*)(t)\mathrm{d}t + \int_0^1 x^*(t)\log\frac{x^*}{x}(t)\mathrm{d}t\right] \\
&\leq 2\nu \cdot \left[\int_0^1 (x - x^*)(t)\mathrm{d}t + \int_0^1 x^*(t)\left(\frac{x^*}{x}(t) - 1\right)\mathrm{d}t\right] \\
&= 2\nu\int_0^1 x(t)\left(\frac{x^*}{x}(t) - 1\right)^2 \mathrm{d}t \\
&= 2\nu\int_0^1 \frac{1}{x(t)}(x^*(t) - x(t))^2\mathrm{d}t \\
&\leq \int_0^1 (x^*(t) - x(t))^2\mathrm{d}t.
\end{aligned}
$$

# K   Implementation Details and Further Numerical Results for Learning Intensity Functions of Poisson Processes with Pseudo Mirror Descent

In this section, we specify the implementation details on pseudo mirror descent and benchmark algorithms. We also provide more detailed experiment results which we did not put in the main text due to space constraints.

## K.1   Learning the Intensity Function of a One-Dimensional Poisson Process

We provide additional experiment comparing pseudo mirror descent, projected gradient descent, and the link function approach. We first introduce the implementation details of PGD and the link function approaches, and then present additional simulation results.

**Projected gradient descent (PGD).** For a one-dimensional Poisson process, we consider a parametric setting. Let $\mathbf{t} := \{t_1, \ldots, t_M\} \subset [0, 1]$ be a set of beacons, and let $\mathbf{c}$ be the coefficient vector. We perform projected gradient descent on the loss function $f$ as a function of $\mathbf{c}$. Similar to the procedure adopted in Yang et al. [2017], we use discretization to avoid computing the integral of the kernel. It has been shown in Yang et al. [2017] that the approximation error brought about can be well-controlled. In particular, we approximate

$$\int_0^z K(x, y)\mathrm{d}x \approx \sum_{k=1}^{\lfloor z/\Delta \rfloor} \Delta K(k \cdot \Delta, y).$$

where the right-hand side is the Riemann sum of the integral on the left-hand side with $\Delta$ being the length of the interval. Let $B$ denote the size of the mini-batch and $t_{ij}$ denoting the time of the $j$-th arrival in the $i$-th sample path in the mini-batch, we have

$$\nabla f(\mathbf{c}) = \sum_{k=1}^{\lfloor z/\Delta \rfloor} \Delta K(k \cdot \Delta, \cdot) - \frac{1}{B} \sum_{i=1}^{B} \sum_{j=1}^{m_i} f^{-1}(k \cdot \Delta) K(t_{ij}, \cdot),$$

where $m_i$ is the number of arrivals in the $i$-th sample path. This allows us to perform updates via

$$\mathbf{c}^{(k+1/2)} = \mathbf{c}^{(k)} - \eta_k \nabla f(\mathbf{c}).$$

Notice that the update is not complete and we need to obtain $\mathbf{c}^{(k+1)}$ by projecting $\mathbf{c}^{(k)}$ so that $x^{(k+1)}$ is positive. To perform the projection, we aim to find the coefficient vector $\mathbf{c}^{(k+1)}$ that minimizes $\|K(\cdot, \mathbf{t})\mathbf{c}^{(k+1/2)} - K(\cdot, \mathbf{t})\mathbf{c}^{(k+1)}\|_2$ while satisfying $K(\cdot, \mathbf{t})\mathbf{c}^{(k+1)} \geq 0$. To simplify computation, we solve the following problem in practice:

$$\underset{\mathbf{c}^{(k+1)} \in \mathbb{R}^{k+1}}{\text{minimize.}} \left( \mathbf{c}^{(k+1)} - \mathbf{c}^{(k+1/2)} \right)^\top \mathbf{K} \left( \mathbf{c}^{(k+1)} - \mathbf{c}^{(k+1/2)} \right), \qquad \text{s.t.} \quad \mathbf{K}\mathbf{c}^{(k+1)} \geq 0,$$

where we have used $\mathbf{K} \in \mathbb{R}^{M \times M}$ to denote a Gram matrix whose $i, j$-th entry $\mathbf{K}_{ij} = K(t_i, t_j)$. This reduces to minimize the distance of the function represented by $\mathbf{c}^{(k+1/2)}$ and the projected function represented by $\mathbf{c}^{(k+1)}$ at the beacons $t_1, \ldots, t_M$, subject to positivity constraints only at those beacons. This requires solving a quadratic programming problem which we solve using cvxopt [Andersen et al., 2013].

**RKHS + Link function $x = y^2$.** We adopt and revise the method in Yang et al. [2017] and apply it to the Poisson process. The original algorithm proposed in Yang et al. [2017] was designed to perform online estimation for the triggering functions of the multivariate Hawkes process (MHP). It can also be used for estimating the intensity of Poisson counting processes in the batch learning setting. Upon assuming that the intensity function is $x(t) = y^2(t)$, we immediately have the alternative form of the loss function:

$$f(y) = \int_0^1 y^2(t) - x^*(t) \log y^2(t) \mathrm{d}t.$$

The gradient is then

$$\nabla f(y) = \int_0^1 2y(t)K(t, \cdot)\mathrm{d}t - \frac{1}{B} \sum_{i=1}^{B} \sum_{j=1}^{m_i} \frac{2K(t_{ij}, \cdot)}{y(t_{ij})}$$

$$\approx \sum_{k=1}^{\lfloor z/\Delta \rfloor} \Delta K(k \cdot \Delta, \cdot) - \frac{1}{B} \sum_{i=1}^{B} \sum_{j=1}^{m_i} \frac{2K(t_{ij}, \cdot)}{y(t_{ij})}.$$

**Simulation settings and results.** We adopt the simulation settings in Table 1. We show the estimation results of the pseudo mirror descent under the step sizes specified in Figure 1 in Figure 6.

| Method | Initialization | $\eta_k$ | $K(x, y)$ | Solver |
|---|---|---|---|---|
| PMD (Fig. 1) | $x^{(0)} \equiv 10$ | See Figure 1 | $(1 + xy)^2$ | None |
| PMD (Fig. 1) | $x^{(0)} \equiv 10$ | $(0.1k + 10)^{-1}$ | $(1 + xy)^2$ | None |
| PGD (Fig. 1) | $\mathbf{c}^{(0)} \equiv 0.1$ | $(2k + 100)^{-1}$ | $\exp(-(x - y)^2/2)$ | cvxopt |
| NPOLE (Fig. 1) | $x^{(0)} \equiv 10$ | $(k + 100)^{-1}$ | $\exp(-(x - y)^2/2)$ | None |

Table 1: Simulation settings for Figure 1. PMD: pseudo mirror descent; PGD: projected gradient descent; NPOLE: RKHS + link function approach.

Figure 6: Learning result for pseudo mirror descent with constant (left) and vanishing (right) step sizes.

## K.2 Learning Shot Distances in Professional Basketball Games

The simulation setting for the experiment in Figure 3 is given in Tables 2, 3 and 4.

Apart from the pseudo mirror descent and the link function approaches, we also built a neural network estimator as a competitive benchmark. The neural network we used is fully connected and has 3 hidden layers of size 64. The activation function for hidden layers is the hyperbolic tangent function. The activation function for the output layer is $\sigma(t) = tanh(t^2)$, to guarantee positivity and boundedness of the output.

| Method | Initialization | $\eta_k$ | $K(x,y)$ | Solver |
|--------|----------------|----------|----------|--------|
| PMD | $x^{(0)} \equiv 1$ | $0.01$ | $1 + \min\{x,y\}$ | None |
| NPOLE | $x^{(0)} \equiv 1$ | $(0.1\sqrt{k} + 10)^{-1}$ | $\exp(-(x-y)^2/(2 \times 0.1^2))$ | None |

Table 2: Simulation settings for Figure 3 on Stephen Curry.

| Method | Initialization | $\eta_k$ | $K(x,y)$ | Solver |
|--------|----------------|----------|----------|--------|
| PMD | $x^{(0)} \equiv 1$ | $(10\sqrt{k} + 100)^{-1}$ | $1 + \min\{x,y\}$ | None |
| NPOLE | $x^{(0)} \equiv 1$ | $(10\sqrt{k} + 100)^{-1}$ | $\exp(-(x-y)^2/(2 \times 0.1^2))$ | None |

Table 3: Simulation settings for Figure 3 on Klay Thompson.

| Method | Initialization | $\eta_k$ | $K(x,y)$ | Solver |
|--------|----------------|----------|----------|--------|
| PMD | $x^{(0)} \equiv 1$ | $0.01$ | $1 + \min\{x,y\}$ | None |
| NPOLE | $x^{(0)} \equiv 1$ | $(0.1\sqrt{k} + 10)^{-1}$ | $\exp(-(x-y)^2/(2 \times 0.1^2))$ | None |

Table 4: Simulation settings for Lebron James in Figure 3.

## L Online Learning for Multivariate Hawkes Process

The pseudo mirror descent algorithm can also be applied successfully to online learning scenarios. In the experiment below, we extend the NPOLE-MHP algorithm proposed in Yang et al. [2017] to efficiently learn triggering functions in multivariate Hawkes processes, which we present as a stand-alone algorithm in Algorithm 2. This algorithm requires the input of a sequence of step sizes, as well as a window size $z$, which was required in the NPOLE-MHP for the approximate evaluation of the intensity function. As an online algorithm, Algorithm 2 updates at a set of given intervals $t_0, \ldots, t_{T-1}$, with $t_0 = 0$ and $t_{k+1}$ being the minimum of $t_k + \delta$, with $\delta$ being a small but fixed time interval, and the arrival time of the next event. We denote $I_{ik}$ as an indicator function indicating whether there is an arrival on the $i$-th dimension during the interval $[t_k, t_{k+1})$. We can

see that Algorithm 2 holds a similar structure to the NPOLE-MHP. However, unlike NPOLE-MHP, the algorithm *does not* require performing projections any more, and only requires computing the multiplicative update rule with a pseudo-gradient evaluated in (12). Rigorous proof of convergence is out of the scope of this paper, and instead we demonstrate the performance of Algorithm 2 on both synthetic and real datasets.

---

**Algorithm 2** NonParametric OnLine Estimation for MHP (NPOLE-MHP) with pseudo mirror descent

---

1: **input:** a sequence of step sizes $\{\eta_k\}_{k=1}^{\infty}$, window size $z$.
2: Initialize $y_{ij}^{(0)}$ for all $i, j$.
3: **for** $k = 0, ..., T - 1$ **do**
4:     Observe the interval $[t_k, t_{k+1})$, and compute $I_{ik}$ for $i \in \{1, \ldots, p\}$.
5:     **for** $i = 1, \ldots, p$ **do**
6:         Evaluate pseudo-gradient $g_{ij}^{(k+1)}$ according to (12) for all $j$.
7:         **for** $j = 1, \ldots, p$ **do**
8:             Set $y_{ij}^{(k+1)} \leftarrow y_{ij}^{(k)} \exp(-\eta_k g_{ij}^{(k+1)})$.
9:         **end for**
10:     **end for**
11: **end for**
12: **output:** $y_{ij}^{(T)}$ for all $i, j$.

---

### L.1   A 5-Dimensional Synthetic Dataset

**Simulation setup.** We used the synthetic data set in Yang et al. [2017], where the triggering function matrix is

$$
\mathbf{Y} = \begin{bmatrix}
e^{-2.5t} & 0 & 0 & e^{-10(t-1)^2} & 0 \\
2^{-5t} & (1 + \cos(\pi t))e^{-t}/2 & e^{-5t} & 0 & 0 \\
0 & 2e^{-3t} & 0 & 0 & 0 \\
0 & 0 & 0 & e^{-2t^2} & e^{-4t} \\
0 & 0 & te^{-5(t-1)^2} & 0 & e^{-3t}
\end{bmatrix}.
$$

We simulated pseudo mirror descent, with initialization being the same as the setting in Yang et al. [2017], and $\eta_k$ being $1/(k * 0.00001 + 1)$, $1/(k * 0.00001 + 0.5)$, $1/(k * 0.000005 + 1)$, $1/(k * 0.00001 + 1)$, $1/(k * 0.00001 + 1)$, accordingly, for the 5 dimensions. We changed the update rule to pseudo mirror descent update with the pseudo-gradient calculated at each iteration by (12) with a Sobolev kernel $K(x, y) = 1 + \min\{x, y\}$, and adjusted the step sizes accordingly.

For benchmarks, we used NPOLE-MHP results, recreated from the settings in Yang et al. [2017]. We also built a neural network estimator as a benchmark. The neural network we used is fully connected and has 3 hidden layers of size 16. The activation functions for hidden layers are rectified linear units (ReLU). The activation function for the output layer is $\sigma(t) = \tanh(t^2)$, to guarantee positivity and boundedness of the output.

**Results.** We first selected 3 out of 25 triggering functions to display in Figure 7, which showed that pseudo mirror descent achieved a similar accuracy to the benchmarks. We note that the bumps in the curves in the second subfigure is due to the fact that we used a nondifferentiable kernel and pointwise evaluation of the triggering and intensity functions. This motivates us to simulate a set of parallel results using a smoother Gaussian kernel, which we present subsequently.

A more complete simulation result is given in Figure 8. We can see that pseudo mirror descent performs similar to NPOLE-MHP on estimating non-zero $y_{ij}$'s. Although the result is similar to that of the neural network, it is partially due to the fact that we gave the neural network extra information during its training phase. This includes the dimensions for which $y_{ij}$ is non-zero, and the exact values

Figure 7: Comparison between pseudo mirror descent, NPOLE-MHP (online gradient descent), and neural networks on a 5-dimensional multivariate Hawkes process.

for the base intensities. Even so, we found it very hard to train the neural network, which often converges to solutions far from optimal.

Lastly, for the synthetic dataset, we compared the performance of pseudo mirror descent using different kernels in Figure 9. In particular, we used a Sobolev kernel, $K(x, y) = 1 + \min\{x, y\}$, and a Gaussina kernel, $K(x, y) = \exp(-(x - y)^2/0.02)$. The result shows that, under pointwise evaluation, the Gaussian kernel creates a smoother output.

## L.2  A Memetracker Dataset

As our last additional experiment, we compared the performance of pseudo mirror descent (Algorithm 2), NPOLE-MHP, and MLE-SGLP [Xu et al., 2016] on a memetracker dataset [Leskovec et al., 2009]. Following the settings of Yang et al. [2017], we collected the publication times of posts and articles of the 20 most active websites, and studied the pattern of their publication behaviors using 10 days worth of data. We modeled the process as a 20-dimensional Hawkes process. The result is reported in Figure 10, where we plotted a $20 \times 20$ heatmap with each cell being the $\mathcal{L}_1$-norm of a triggering function. We can see that pseudo mirror descent generates a similar heat map compared to NPOLE-MHP and MLE-SGLP, showing mainly self-exciting behavior of the dataset.

Figure 8: Performance comparison between pseudo mirror descent (PMD) with Sobolev kernel, NPOLE-MHP, and the neural network on estimating $\mathbf{Y}$. The black curve is the ground truth, the green curve is the result of NPOLE-MHP. The red curve is the result of PMD, and the orange curve is the result of the neural network. Due to the difficulty in training neural network to obtain a comparable result, we gave it extra information of the base intensity $x_{i0}^*$'s, as well as the information regarding which function is non-zero.

Figure 9: Performance comparison between pseudo mirror descent using the Sobolev kernel (red), and the Gaussian kernel (orange), compared with the ground truth (black).

Figure 10: A heatmap comparison between the estimates of pseudo mirror descent, NPOLE-MHP, and MLE-SGLP on the memetracker dataset.