[Reviews · NeurIPS 2019]

Reviewer 1



There are various applications in ML (e.g. intensities in point processes) that learning a positive function is desired. Link function and Projection-based methods are among the popular approaches to this problem. This paper proposes a nonparametric efficient method with the theoretical guarantee to learn such functions. How to choose the proper Bregman divergence? In the eq(4), what is the role of the first term which does not depend on x? The paper is generally well written and the conditions are clearly mentioned. The impact of the work is justified by the authors. I generally liked the idea and recognized it as a good step towards constrained optimization. Update: After reading the rebuttal, I would like to keep my score and recommend this paper for acceptance.

Reviewer 2



Originality: the paper proposes a pseudo-mirror descent framework that can be applied to handle positive-value function learning problems such as Poisson processes and Hawkes processes. Related RKHS formulation of learning these processes exist in a batch-learning setting. Quality and clarity: The paper provides both theoretical justifications as well as real-world examples to demonstrate the utility of the proposed method. The presentation of the paper is also clear. Significance: In section 3.1, the intensity function is assumed to be continuous. The reviewer is concerned with whether such an assumption could be restrictive as in many scenarios modeled by point processes the intensity function can demonstrate sudden changes and hence is not continuous. Overall, the author demonstrates a nice way to carry out optimization in Hilbert space and offer solutions to interesting point process problems. The convergence results of the psudo mirror descent algorithm can also be of interest to non-convex optimization. =========== after feedback ========== The reviewer would like to thank the authors for their explanation. In terms of the concern in continuity, a nice addition would be some synthetic experiments where there are sudden changes in the intensity functions (e.g. how well the method can track a step function of intensity).

Reviewer 3



pros: 1) This paper is well-written and easy to follow. 2) Theoretical assumptions for analysis and definitions are very clear. 3) Concrete examples are given to guide readers to interpret the definitions. cons: 1) The implication of some important theoretical results (e.g., theorem 3 and corollary 4) are not well explained. 2) The experimental results are not well analyzed. It seems the proposed algorithm tend to oscillate after the first few iterations compared with the link function approach and projected gradient descent. It would be better that the authors could analyze this in detail to show some hidden properties of the new algorithm. 3) The experimental settings are a bit confusing. The reviewer usually chooses the same kernel across different experiments for convenience to justify the robustness of the proposed method. However, the authors used different kernels in synthetic and real experiments. It would be better to briefly explain the rationale behind.

[Author Response · NeurIPS 2019]

We thank all reviewers for their comments and acknowledgement of our contribution. All comments are very useful and will be addressed in greater details in the revision. For the theory part, we will add discussions on key results such as Theorem 3 and Corollary 4, as Reviewer 3 suggested. For the experiments, we will add (i) more detailed illustrations and analysis to the experimental settings and results, (ii) insights on the trade-offs of using different types of kernels, as well as (iii) more comprehensive comparisons between pseudo mirror descent and the existing benchmarks. Below we address each reviewer's comments separately.

**Response to Reviewer 1**:

**How to choose the proper Bregman divergence?** The choice of Bregman divergence is rather flexible and problem-specific depending on the underlying geometry. In our context of learning positive functions, any distance-generating function $\Phi$ such that $\nabla\Phi^*$ ensures positivity would be sufficient. This includes the entropy function $\Phi(x) = \int x(t)\log x(t)\mathrm{d}t - \int x(t)\mathrm{d}t$ (resp. the generalized I-divergence), the negative logarithmic function $\Phi(x) = -\int \log x(t)\mathrm{d}t$, (resp. the Itakura-Saito divergence), and simple functions such as $\Phi(x) = \int \frac{2}{5}x^{5/2}(t)\mathrm{d}t$, just to name a few. It is yet unclear whether there exist ways to systematically design the "best Bregman divergence in a theoretical way. Instead, we will provide further numerical comparisons under different Bregman divergences to further illustrate this point in the revision.

**The role of the first term in (4).** While adding this term or not does not affect the update rule, it is presented here only to emphasize the fact that $x^{(k)}$ is obtained by minimizing a 2nd-order Taylor approximation of $f(x)$ at $x^{(k-1)}$, with the quadratic term replaced by the Bregman divergence. This is also commonly adopted in the literature.

**Response to Reviewer 2**:

**Is continuity of the intensity function restrictive?** We think that the continuity of the intensity function is indeed a minimal assumption here for nonparametric estimation. Note that many existing literature assumes even more restrictive smoothness conditions (learning in Sobolev spaces, smoothing spline expansion, etc). Moreover, many real-world data are well captured by continuous intensity functions, as also demonstrated in our experiments. In the case where the true underlying intensity function is discontinuous, our algorithm would return a close continuous approximation. We will further illustrate this point in our revision.

**Elaborate potential limitations?** The only potential limitation of our results we can think of is that the current analysis is not fully generalizable to incorporate additional constraints on the intensity function on top of positivity. In general, the pseudo mirror descent algorithm can be applied to solve constrained problems, but our current analysis is only applicable to enforcing positivity. A potential remedy is that one could convert additional constraints to penalty terms. Closing this gap is an interesting direction we are working on that is not present in the current version of the paper.

**Response to Reviewer 3**:

**Implications of Theorem 3 and Corollary 4.** Thanks for the suggestion. We will work on better explaining these results in the revision to help understanding. In plain words, Theorem 3 says asymptotically, the inner product between pseudo-gradient and the gradient in "dual space" goes to 0. This result becomes particularly useful if we set the pseudo-gradient as in Corollary 4, which then implies asymptotic vanishing of the gradient norm. Depending on specific forms of the objective function, this may further imply that pseudo mirror descent converges to a stationary point.

**Clarification on the oscillation behavior in experiments.** We suppose the reviewer is referring to the right plot of Figure 1. There could be several reasons causing the oscillation: (i) first and foremost, unlike gradient descent, mirror descent is not necessarily a "descent" method (despite its name), i.e., the objective is not necessarily decreasing monotonically, (ii) the estimate is already close to the ground truth, so small noise could also cause oscillation.

**Clarification on the kernel choices in experiments.** We apologize for the confusion. Our original intention is to demonstrate that the algorithm works well under different choices of kernels. There is also an interesting intuition behind which kernel to use that is not shown in the current version of the paper. Using finite-dimensional kernel, such as polynomial kernels, would guarantee (7) and hence the rates in Theorem 6, while using infinite-dimensional kernels, such as the Sobolev kernel, generally has faster convergence at early stage. We will add more discussions on the kernel choices in the revision.

[Meta-Review · NeurIPS 2019]

An overall strong contribution for learning positive functions. The following could be cleaned up - Improved explanation of the importance of the theoretical results - Explaining the choice of kernels - Synthetic experiments with jumps in the intensity function